# Reverting the mode of action of the mitochondrial $F_OF_1$-ATPase by *Legionella pneumophila* preserves its replication niche

**Pedro Escoll[1]\*, Lucien Platon[1,2†‡], Mariatou Dramé[1,3†], Tobias Sahr[1], Silke Schmidt[1,4], Christophe Rusniok[1], Carmen Buchrieser[1]\***

[1]Institut Pasteur, Biologie des Bactéries Intracellulaires and CNRS UMR 3525, Paris, France; [2]Faculté des Sciences, Université de Montpellier, Montpellier, France; [3]Faculté des Sciences, Université de Paris, Paris, France; [4]Sorbonne Université, Collège doctoral, Paris, France

**\*For correspondence:**
pescoll@pasteur.fr (PE);
cbuch@pasteur.fr (CB)

[†]These authors contributed equally to this work

**Present address:** [‡]Institut Pasteur, Génétique du Paludisme et Résistance, Paris, France

**Competing interest:** The authors declare that no competing interests exist.

**Abstract** *Legionella pneumophila,* the causative agent of Legionnaires' disease, a severe pneumonia, injects via a type 4 secretion system (T4SS) more than 300 proteins into macrophages, its main host cell in humans. Certain of these proteins are implicated in reprogramming the metabolism of infected cells by reducing mitochondrial oxidative phosphorylation (OXPHOS) early after infection. Here. we show that despite reduced OXPHOS, the mitochondrial membrane potential ($\Delta\psi_m$) is maintained during infection of primary human monocyte-derived macrophages (hMDMs). We reveal that *L. pneumophila* reverses the ATP-synthase activity of the mitochondrial $F_OF_1$-ATPase to ATP-hydrolase activity in a T4SS-dependent manner, which leads to a conservation of the $\Delta\psi_m$, preserves mitochondrial polarization, and prevents macrophage cell death. Analyses of T4SS effectors known to target mitochondrial functions revealed that *Lp*Spl is partially involved in conserving the $\Delta\psi_m$, but not LncP and MitF. The inhibition of the *L. pneumophila*-induced 'reverse mode' of the $F_OF_1$-ATPase collapsed the $\Delta\psi_m$ and caused cell death in infected cells. Single-cell analyses suggested that bacterial replication occurs preferentially in hMDMs that conserved the $\Delta\psi_m$ and showed delayed cell death. This direct manipulation of the mode of activity of the $F_OF_1$-ATPase is a newly identified feature of *L. pneumophila* allowing to delay host cell death and thereby to preserve the bacterial replication niche during infection.

## Editor's evaluation

The pathogenic bacterium *Legionella pneumophila* (Lp) is known for its ability to translocate cocktails of effector proteins into its eukaryotic host. Yet, despite an overall reduction in mitochondrial oxidative phosphorylation following *Legionella* infection, host cell mitochondria maintain their normal membrane potential ($\Delta\phi_m$). In this study, the authors show that the translocated effector protein LpSpl forces the host cell's FOF1-ATPase to function in 'reverse mode,' thereby maintaining $\Delta\phi_m$, which, ultimately, supports bacterial replication by delaying host death.

## Introduction

Beyond their essential role in cellular bioenergetics, mitochondria are integrated into diverse signaling pathways in eukaryotic cells and perform various signaling functions, such as immune responses or cell death, as they play crucial roles in the regulation of apoptosis (***Bock and Tait, 2020***). Thus,

mitochondria are targeted by several intracellular bacteria during infection to modulate their functions to the bacterial advantage (*Spier et al., 2019*). One of these bacteria is *Legionella pneumophila*, the causative agent of Legionnaires' disease. We have shown previously that this pathogen targets mitochondrial dynamics during infection of primary human monocyte-derived macrophages (hMDMs) by injecting type 4 secretion system (T4SS) effectors such as MitF, leading to a fragmented mitochondrial network via the recruitment of the host fission protein DNM1L to the mitochondrial surface (*Escoll et al., 2017b*). Importantly, *Legionella* induced mitochondrial fragmentation at early time points such as 5 hr post-infection (hpi), when bacterial replication has not started yet, and in the absence of cell death signs. The fragmentation of mitochondrial networks provoked a T4SS-dependent reduction of mitochondrial respiration in *Legionella*-infected macrophages, evidencing a functional connection between mitochondrial dynamics and mitochondrial respiration (*Escoll et al., 2017b*).

Mitochondrial respiration results from coupling the activity of five complexes in the electron transport chain (ETC) at mitochondrial cristae. In this process, the reduced coenzymes NADH and $FADH_2$ generated at the mitochondrial matrix by the tricarboxylic acid (TCA) cycle are oxidized at complexes I and II where their electrons are extracted to energize the mitochondrial ETC (*Nolfi-Donegan et al., 2020*). The sequential transit of these electrons through complexes I, III, and IV allows to pump protons from the matrix to the intermembrane space (IMS) and at complex IV, diatomic oxygen $O_2$ serves as the terminal electron acceptor, and $H_2O$ is formed. The increased concentration of protons $[H^+]$ at the IMS, compared to $[H^+]$ at the matrix, generates the mitochondrial membrane potential ($\Delta\psi_m$). This is necessary to produce ATP by fueling the rotation of complex V, the mitochondrial $F_OF_1$-ATPase, in a process termed oxidative phosphorylation (OXPHOS) (*Nolfi-Donegan et al., 2020*). Our previous studies determined that at 5 hpi *L. pneumophila*, by altering mitochondrial dynamics, reduced OXPHOS as well as the cellular ATP content in hMDMs in a T4SS-dependent manner (*Escoll et al., 2017b*).

Why *L. pneumophila* and other species of intracellular bacteria reduce mitochondrial OXPHOS during infection of host cells remains a matter of debate (*Escoll and Buchrieser, 2018*; *Russell et al., 2019*). As intracellular bacteria can obtain resources only from host cells, it has been suggested that halting mitochondrial OXPHOS during infection might benefit pathogenic bacteria by redirecting cellular resources, such as glycolytic or TCA intermediates, to biosynthetic pathways that might sustain intracellular bacterial replication instead of fueling mitochondria (*Escoll and Buchrieser, 2018*; *Russell et al., 2019*). For instance, it has been shown that *Mycobacterium tuberculosis* redirects pyruvate to fatty acid synthesis and *Chlamydia trachomatis* subverts the pentose phosphate pathway to increase the synthesis of nucleotides for its own intracellular growth (*Siegl et al., 2014*; *Singh et al., 2012*). On the other hand, upon sensing bacterial lipopolysaccharides, macrophages redirect mitochondrial TCA intermediates, such as citrate or succinate, to drive specific immune functions such as the production of cytokines or the generation of antimicrobial molecules (*Escoll and Buchrieser, 2019*; *Russell et al., 2019*; *O'Neill and Pearce, 2016*). Thus, while these metabolic shifts, which are redirecting resources from mitochondria to the cytoplasm, should be activated in macrophages to develop their antimicrobial functions, they could also benefit intracellular bacteria as more resources would be available in the cytoplasm for bacterial growth. Importantly, reduction of OXPHOS may lead to decreased $\Delta\psi_m$ and ATP production at mitochondria, which are events that trigger the activation of cell death programs. How intracellular bacteria withdraw OXPHOS, deal with the subsequent $\Delta\psi_m$ drop, and host cell death but manage to preserve their host cell to conserve their replication niche is a question that remains poorly understood.

To answer this question, we monitored the evolution of mitochondrial polarization during infection of hMDMs by *L. pneumophila* and showed that in the absence of OXPHOS *L. pneumophila* regulates the enzymatic activity of the mitochondrial $F_OF_1$-ATPase during infection. This allows maintaining the $\Delta\psi_m$ and delays cell death of infected hMDMs in a T4SS-dependent manner. Our results identified a new virulence mechanism of *L. pneumophila*, namely, the manipulation of the mitochondrial $F_OF_1$-ATPase to preserve the integrity of infected host cells and thereby the maintenance of the bacterial replication niches.

## Results

### Despite *L. pneumophila*-induced reduction of mitochondrial respiration, the mitochondrial membrane potential is maintained

We have previously shown that *L. pneumophila* strain Philadelphia JR32 impairs mitochondrial respiration during infection (*Escoll et al., 2017b*). Here, we analyzed *L. pneumophila* strain Paris (Lpp) to learn whether this is a general characteristic of *L. pneumophila* infection. We infected hMDMs with Lpp WT or a T4SS-deficient mutant (Δ*dotA*) for 6 hr and analyzed their mitochondrial function compared to uninfected hMDMs by using a cellular *respiratory control assay* in living cells (*Brand and Nicholls, 2011*; *Connolly et al., 2018*). This assay determines oxygen consumption rate (OCR) in basal conditions and during the sequential addition of mitochondrial respiratory inhibitors. OCR variations observed indicate how mitochondrial respiration is functioning in a cell population (*Figure 1A*, *Figure 1—figure supplement 1A*). Our results showed that basal respiration is significantly reduced (p<0.0001) in WT-infected hMDMs compared to Δ*dotA*- and noninfected hMDMs (*Figure 1A and B*). This indicates that $O_2$ consumption, which is predominantly driven by ATP turnover and the flow of $H^+$ to the matrix through the mitochondrial $F_OF_1$-ATPase, is severely impaired in WT-infected hMDMs. Further analysis of OCR changes upon addition of oligomycin, an inhibitor of the mitochondrial $F_OF_1$-ATPase, indicated that the rate of mitochondrial respiration coupled to ATP synthesis is highly reduced in WT-infected hMDMs compared to Δ*dotA*- or noninfected cells. Other respiratory parameters such as proton leak were also reduced in WT-infected macrophages (*Figure 1A*, *Figure 1—figure supplement 1A and B*). Subsequent addition of an uncoupler to create a $H^+$ short-circuit across the inner mitochondrial membrane (IMM), such as FCCP, allowed measuring the maximum respiration rate and the spare respiratory capacity, revealing that both were severely impaired in WT-infected cells compared to Δ*dotA*- and noninfected hMDMs (*Figure 1A*, *Figure 1—figure supplement 1A and B*). Finally, inhibition of the respiratory complexes I and III with rotenone and antimycin A, respectively, measured $O_2$ consumption driven by nonmitochondrial processes, such as cytoplasmic NAD(P)H oxidases, which showed similar levels of nonmitochondrial $O_2$ consumption in all infection conditions (*Figure 1A*, *Figure 1—figure supplement 1A and B*).

Taken together, our results indicated that several mitochondrial respiration parameters were severely altered during infection with Lpp-WT, including respiration coupled to ATP production. Importantly, some of the respiratory parameters measured that are oligomycin-sensitive were reduced in Lpp-WT-infected hMDMs but not in Δ*dotA*-infected cells, suggesting that the mitochondrial $F_OF_1$-ATPase activity may be altered during *L. pneumophila* infection in a T4SS-dependent manner.

The transition of electrons across mitochondrial ETC complexes allows the extrusion of $H^+$ from the matrix to the IMS generating a $H^+$ circuit where the mitochondrial $F_OF_1$-ATPase is the dominant $H^+$ re-entry site during active ATP synthesis by OXPHOS. In cellular steady-state conditions, extrusion and re-entry $H^+$ fluxes across mitochondrial membranes are balanced (*Brand and Nicholls, 2011*). Therefore, any exogenous alteration of ATP turnover and/or $F_OF_1$-ATPase activity influences this $H^+$ circuit and might be reflected in $\Delta\psi_m$ levels. Thus, we decided to quantify the $\Delta\psi_m$ in infected cells. We developed a miniaturized *high-content* assay based on kinetic measurements of tetramethylrhodamine methyl ester (TMRM) fluorescence in nonquenching conditions (10 nM), where TMRM fluorescence in mitochondria is proportional to the $\Delta\psi_m$ (*Connolly et al., 2018*; *Duchen et al., 2003*). This assay allowed to measure changes in the $\Delta\psi_m$ at the single-cell level and in thousands of living cells during infection (*Figure 1C*). Image analysis showed that the $\Delta\psi_m$ slightly increased in Lpp-WT-, Lpp-Δ*dotA*-, and noninfected cell populations during the first hours of infection (1–3 hpi), and progressively decreased during the time course with no differences between the infection conditions (*Figure 1D*). Single-cell analyses (*Figure 1E*, *Figure 1—figure supplement 1C*) showed that Lpp-WT-, Lpp-Δ*dotA*-, and noninfected single hMDMs showed a wide range of $\Delta\psi_m$ values at any time point (*Figure 1—figure supplement 1C*) with no significant differences between Lpp-WT- and Lpp-Δ*dotA*-infected hMDMs at 6 hpi. Infected cells with both strains showed a significantly higher $\Delta\psi_m$ compared to noninfected cells (p<0.0001, *Figure 1E*, *Figure 1—figure supplement 1D*). Thus, despite a significant reduction of OXPHOS in WT-infected cells, the $\Delta\psi_m$ was maintained. This suggests that *L. pneumophila* manipulates the mitochondrial ETC to conserve the $\Delta\psi_m$ of hMDMs in the absence of OXPHOS.

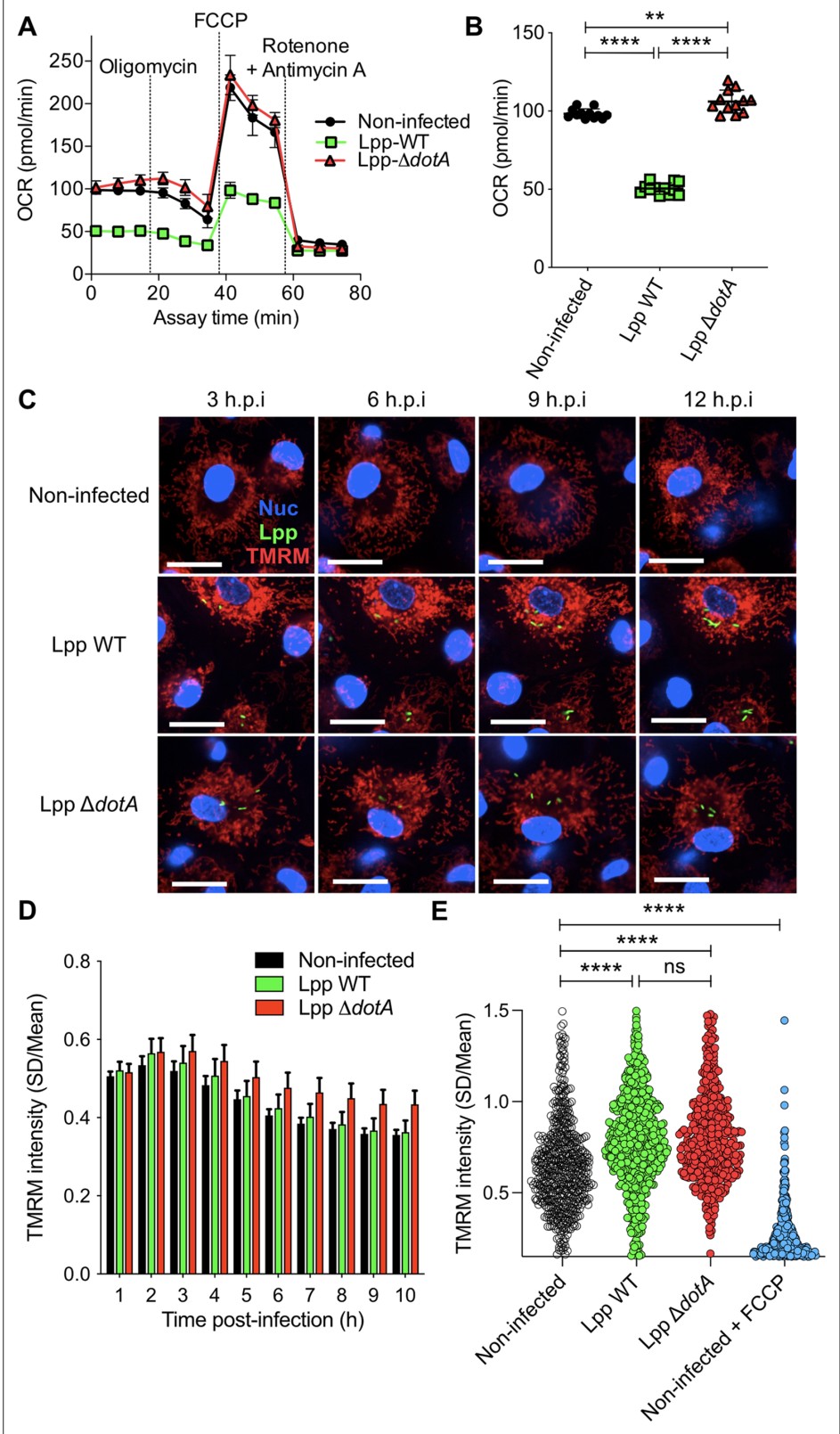

**Figure 1.** Despite a reduction of oxidative phosphorylation (OXPHOS), human monocyte-derived macrophages (hMDMs) maintain their $\Delta\psi_m$ during infection by *L. pneumophila*. (**A**) hMDMs were infected with *L. pneumophila* strain Paris (Lpp) wild-type (WT), a type 4 secretion system (T4SS)-deficient Δ*dotA* mutant, or left uninfected (noninfected). At 6 hr post-infection (hpi), a cellular respiratory control assay was performed by measuring

*Figure 1 continued on next page*

*Figure 1 continued*

oxygen consumption rate (OCR) during the sequential addition of mitochondrial respiratory inhibitors (see also *Figure 1—figure supplement S1A*). (**B**) Basal respiration of hMDMs in the same conditions as in (**A**), at 6 hpi. (**C**) hMDMs were infected as in (**A**) with GFP-expressing bacteria (green), nuclei of host cells were stained with Hoechst (Nuc, blue), and $\Delta\psi_m$ was monitored from 1 to 12 hpi using tetramethylrhodamine methyl ester (TMRM) dye in nonquenching conditions (10 nM). Representative confocal microscope images of noninfected and infected cells at 3, 6, 9, and 12 hpi are shown. Intracellular bacterial replication can be observed in Lpp-WT infected hMDMs at 12 hpi. Bar: 20 µm. (**D**) Quantification of TMRM intensity at 1–10 hpi (expressed as SD/mean) in the assays described in (**C**). Data from four independent experiments with a total of 10 replicates. Single-cell data from the entire time course are shown in *Figure 1—figure supplement S1C*. (**E**) hMDMs were infected with GFP-expressing bacteria or left uninfected (noninfected). At 6 hpi, nuclei of host cells were stained with Hoechst and $\Delta\psi_m$ was measured using TMRM dye in nonquenching conditions (10 nM), and FCCP (10 µM) was added or not to noninfected cells as a control to monitor complete mitochondrial depolarization. Single-cell analysis from assays performed in three donors is shown in *Figure 1—figure supplement S1D*. \*\*p<0.01; \*\*\*\*p<0.00001; ns, nonsignificant (Mann–Whitney *U* test).

The online version of this article includes the following figure supplement(s) for figure 1:

**Figure supplement 1.** Oxidative phosphorylation (OXPHOS) and $\Delta\psi_m$ during infection by *L. pneumophila*.

## *L. pneumophila* infection induces the 'reverse mode' of the mitochondrial $F_O F_1$-ATPase in a T4SS-dependent manner

The mitochondrial $F_O F_1$-ATPase is a fascinating molecular machine that rotates clockwise when it works in the 'forward mode,' synthesizing ATP by using the $\Delta\psi_m$ generated by the $H^+$ circuit (*Figure 2A*, left). It can also rotate counterclockwise when it works in the 'reverse mode' (*Campanella et al., 2009*). In this case, it hydrolyzes ATP to maintain $\Delta\psi_m$ in the absence of OXPHOS (*Figure 2A*, right). As our results showed that *L. pneumophila* highly reduced OXPHOS, likely by an alteration of the $F_O F_1$-ATPase activity, while the $\Delta\psi_m$ was conserved, we investigated in which activity mode the $F_O F_1$-ATPase worked during *Legionella* infection. A widely used method to investigate the directionality of the $F_O F_1$-ATPase in intact cells is to monitor changes in $\Delta\psi_m$ after the addition of $F_O F_1$-ATPase inhibitors, such as oligomycin or dicyclohexylcarbodiimide (DCCD) (*Connolly et al., 2018*; *Gandhi et al., 2009*). These inhibitors block both modes of function; thus if the $F_O F_1$-ATPase is working in the 'forward mode' the $\Delta\psi_m$ will increase after adding the inhibitor as the inhibition of the $H^+$ flux to the matrix through the ATPase leads to an accumulation of $H^+$ at the IMS (*Figure 2B*, left). If $\Delta\psi_m$ decreases after ATPase inhibition, the $F_O F_1$-ATPase works in the 'reverse mode' since now $H^+$ cannot translocate to the IMS to maintain the $\Delta\psi_m$ (*Figure 2B*, right). Here, we used the aforementioned TMRM high-content assay to monitor the $\Delta\psi_m$ in living hMDMs at 6 hpi when OXPHOS is impaired and $\Delta\psi_m$ is maintained.

First, we recorded a baseline and then added medium as a control. As expected, this did not alter $\Delta\psi_m$ in any infection condition (*Figure 2C and D*). However, the addition of FCCP completely depolarized mitochondria, leading to an abrupt drop of $\Delta\psi_m$ in Lpp-WT-, Lpp-$\Delta dotA$-, and noninfected hMDMs (*Figure 2C and E*), demonstrating that this assay can monitor changes in the $\Delta\psi_m$ simultaneously in hundreds of infected cells. To analyze whether the $F_O F_1$-ATPase worked in the synthase (forward) or the hydrolase (reverse) mode, we added oligomycin (*Figure 2F*) or DCCD (*Figure 2G*) to the infected cells. The $\Delta\psi_m$ increased in noninfected or $\Delta dotA$-infected hMDMs, which suggested that the ATPase worked in the 'forward mode' in these infection conditions. In contrast, the addition of oligomycin to Lpp-WT-infected hMDMs had no effect on the $\Delta\psi_m$ (*Figure 2F*), while addition of DCCD decreased the $\Delta\psi_m$ (*Figure 2G*). Thus, our results indicate that the $F_O F_1$-ATPase worked in the 'forward mode' in noninfected or $\Delta dotA$-infected macrophages, whereas the $F_O F_1$-ATPase worked in the 'reverse mode' during infection of hMDMs with the WT strain. This suggests that the induction of the 'reverse mode' depends on the action of T4SS effector(s).

## The T4SS effector *Lp*Spl participates in the induction of the 'reverse mode' of the mitochondrial $F_O F_1$-ATPase during infection

Among the more than 300 bacterial effectors that *L. pneumophila* injects into host cells through its T4SS (*Mondino et al., 2020*), 3 have been shown to target mitochondrial structures or functions. LncP is a T4SS effector targeted to mitochondria that assembles in the IMM and seems to transport ATP across mitochondrial membranes (*Dolezal et al., 2012*). The effector *Lp*Spl (also known as LegS2) was

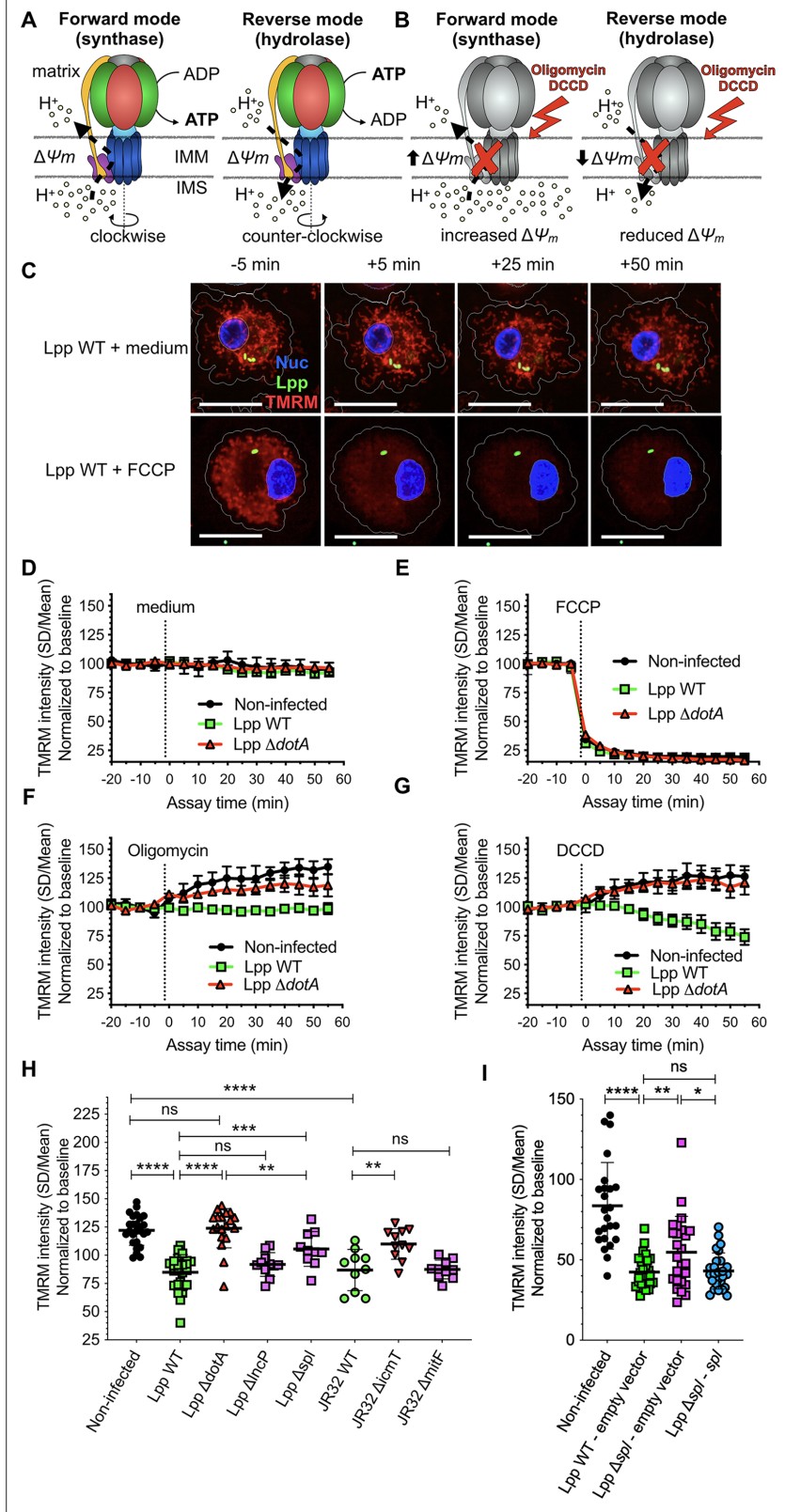

**Figure 2.** The mitochondrial $F_O F_1$-ATPase works in the 'reverse mode' during infection of human monocyte-derived macrophages (hMDMs) by *L. pneumophila*. (**A**) In the 'forward mode' of the mitochondrial ATPase, the $\Delta\psi_m$ generated by the electron transport chain is used by the $F_O F_1$-ATPase to synthesize ATP. The 'reverse mode' of the $F_O F_1$-ATPase leads to ATP hydrolysis to pump $H^+$ to the intermembrane space (IMS). IMM, inner mitochondrial

*Figure 2 continued on next page*

*Figure 2 continued*

membrane. (**B**) When the $F_OF_1$-ATPase is inhibited by oligomycin or dicyclohexylcarbodiimide (DCCD), an increase in $\Delta\psi_m$ indicates that the ATPase was working in the 'forward mode' ($H^+$ accumulate in the IMS), while a decrease in $\Delta\psi_m$ indicates functioning in the 'reverse mode' ($H^+$ cannot be translocated to the IMS by the $F_OF_1$-ATPase to sustain the $\Delta\psi_m$). (**C**) hMDMs were infected with GFP-expressing bacteria (green) or left uninfected (noninfected). At 5.5 hr post-infection (hpi), cells were labeled with Hoechst to identify the cell nucleus (Nuc, blue) and tetramethylrhodamine methyl ester (TMRM) (red) to quantify $\Delta\psi_m$. At 6 hpi, addition of medium (no changes) or FCCP (complete depolarization) was used as controls. Representative confocal images of Lpp-WT-infected hMDMs (6 hpi) at 5 min before the addition of medium (top) or FCCP (bottom), and at 5, 25, and 50 min after addition of medium or FCCP. Bar: 20 μm. (**D**) Quantification of (**C**) before (baseline) and after the addition of medium. Each dot represents mean ± SD of three independent experiments with a total of eight replicates. (**E**) Same as (**D**) but FCCP was added. (**F**) Same as (**D**) but oligomycin was added. (**G**) Same as (**D**) but DCCD was added. (**H**) Same as (**C**) but infection was performed with Lpp-WT, Lpp-ΔdotA, Lpp-ΔlncP, Lpp-Δspl, *L. pneumophila* strain Philadelphia JR32 (JR32)-WT, JR32-ΔicmT or JR32-ΔmitF. TMRM values (SD/mean) at 50 min after DCCD addition are shown. Data from a minimum of three experiments per strain with 10 or more replicates per strain. (**I**) hMDMs were infected with GFP-expressing bacteria or left uninfected (noninfected). At 5.5 hpi, cells were labeled with Hoechst to identify the cell nucleus and TMRM to quantify $\Delta\psi_m$. At 6 hpi, addition of DCCD revealed whether the $F_OF_1$-ATPase works in the 'reverse mode.' Infection was performed with Lpp-WT expressing empty pBCKS vector (empty vector), Lpp-Δspl-expressing empty vector and Lpp-Δspl-expressing pBCKS-spl vector, which express *Lp*Spl (complemented strain, Lpp-Δspl:: spl). Data from three donors are shown. Each dot represents a replicate. *p<0.1; **p<0.01; ***p<0.001; ****p<0.00001; ns, nonsignificant (Mann–Whitney *U* test).

The online version of this article includes the following figure supplement(s) for figure 2:

**Figure supplement 1.** Basal respiration of human monocyte-derived macrophages (hMDMs) infected with *L. pneumophila* Δspl mutant and validation of the model in HEK-293 cells.

suggested to target mitochondria (**Degtyar et al., 2009**), the endoplasmic reticulum (ER) (**Rolando et al., 2016**), and mitochondrial-associated membranes (MAMs) (**Escoll et al., 2017a**). *Lp*Spl encodes a sphingosine-1 phosphate (S1P) lyase that directly targets the host sphingolipid metabolism and restrains autophagy in infected cells. MitF (LegG1) activates the host small GTPase Ran to promote mitochondrial fragmentation during infection of human macrophages (**Escoll et al., 2017b**).

To learn if any of these effectors is involved in the T4SS-dependent induction of the 'reverse mode' of the $F_OF_1$-ATPase, we infected hMDMs for 6 hr with Lpp-WT or its isogenic mutants lacking a functional T4SS (Lpp-ΔdotA), lacking the effector LncP (Lpp-ΔlncP), lacking the effector *Lp*Spl (Lpp-Δspl), and *L. pneumophila* strain Philadelphia JR32 (JR32-WT) and its isogenic mutants lacking the T4SS (JR32-ΔicmT) or the effector MitF (JR32-ΔmitF). Using the TMRM high-content assay, we measured the $\Delta\psi_m$ after the inhibition of the $F_OF_1$-ATPase by DCCD (*Figure 2H*). Our results indicated that, while the $F_O$-$F_1$-ATPase worked in the 'forward mode' in noninfected hMDMs and during infection with T4SS-deficient mutants (Lpp-ΔdotA and JR32-ΔicmT), the $F_OF_1$-ATPase worked in the 'reverse mode' during infection with the Lpp-WT and JR32-WT strains (*Figure 2G and H*). As infection of hMDMs by *L. pneumophila* strain Paris and *L. pneumophila* strain Philadelphia JR32 showed similar results, this suggests that inhibition of mitochondrial respiration and conservation of $\Delta\psi_m$ through the induction of the 'reverse mode' of the mitochondrial ATPase is a virulent strategy of *L. pneumophila*. Infection with Lpp-ΔlncP and JR32-ΔmitF was not significantly different compared to the WT strains, suggesting that these effectors are not involved in the induction of the 'reverse mode' of the mitochondrial ATPase. However, mitochondria of cells infected with Lpp-Δspl showed a significantly higher $\Delta\psi_m$ after DCCD treatment than mitochondria of cells infected with Lpp-WT (p=0.0006), and a significantly lower $\Delta\psi_m$ after DCCD treatment than mitochondria of cells infected with the Lpp-ΔdotA strain (p=0.0034). To confirm that *Lp*Spl is indeed implicated in the induction of the 'reverse mode' of the $F_OF_1$-ATPase, we complemented the Lpp-Δspl mutant strain with a plasmid expressing *Lp*Spl (Lpp Δspl-spl) and infected hMDMs. Cells infected with the WT strain carrying the empty vector (Lpp WT-empty vector) showed a significantly lower $\Delta\phi_m$ after DCCD treatment than noninfected cells (p<0.0001), and the Lpp Δspl mutant carrying the empty plasmid (Lpp Δspl-empty vector) showed a significantly higher $\Delta\phi_m$ after DCCD treatment compared to WT strain (p=0.0095), as shown above. Complementation of Lpp Δspl recovered the WT phenotype as no significant differences in $\Delta\phi_m$ values after DCCD treatment compared to WT-infected cells were observed, but significant differences were still observed with the Lpp Δspl-empty vector (p=0.0216) mutant strain, confirming a role of *Lp*Spl in the induction

of the 'reverse mode' of the $F_OF_1$-ATPase (*Figure 2I*). We also found a very small but significant difference in the oxygen consumption levels exhibited by Lpp-WT- and Lpp $\Delta spl$-infected hMDMs (p=0.0148, *Figure 2—figure supplement 1A*), suggesting that the role of *Lp*Spl in the induction of the 'reverse mode' to maintain $\Delta\phi_m$ during infection might partially involve the modulation of the functioning of the OXPHOS machinery.

To further analyze our results obtained in hMDMs in an easy to transfect cell model, we used HEK-293 cells stably expressing the FcγRII receptor (*Arasaki and Roy, 2010*) as these cells allow efficient internalization of opsonized *L. pneumophila*. The bacteria were opsonized using a monoclonal antibody targeting *L. pneumophila* flagellin. First, the $\Delta\psi_m$ of noninfected cells was monitored upon addition or not of FCCP, showing that it decreased as expected compared to the addition of medium (*Figure 2—figure supplement 1B and C*). Then, we measured the direction of the $F_O$-$F_1$-ATPase during infection by treating cells with DCCD (like in hMDMs, *Figure 2G*). Our results showed that the $\Delta\psi_m$ decreased upon addition of DCCD in Lpp-WT-infected HEK-293 cells, confirming that the $F_O$-$F_1$-ATPase of WT-infected HEK-293 cells worked in the 'reverse mode' (*Figure 2—figure supplement 1D*). The results obtained for the mutant strains during infection of HEK-293 cells were equivalent to those shown in *Figure 2H* obtained in hMDMs. The change of the $\Delta\psi_m$ upon DCCD addition in $\Delta dotA$-infected cells was similar to those in noninfected cells, while results obtained in $\Delta spl$-infected HEK-293 cells showed an intermediate phenotype between the WT and the $\Delta dotA$ mutant, but significantly different to both strains (*Figure 2—figure supplement 1E and F*).

Together, these data suggest that *Lp*Spl is partially involved in the induction of the 'reverse mode' of the $F_OF_1$-ATPase; however, other additional T4SS effector(s) seem to participate in the modulation of the $F_OF_1$-ATPase activity mode.

## Inhibition of *Legionella*-induced 'reverse mode' collapses the $\Delta\psi_m$ and ignites cell death of infected macrophages

To further analyze the importance of the activity mode of the $F_OF_1$-ATPase during infection, we used BTB06584 (hereafter called BTB), a specific inhibitor of the 'reverse mode' of the mitochondrial $F_OF_1$-ATPase (*Ivanes et al., 2014*). We used the TMRM high-content assay and added BTB to noninfected, Lpp-WT- or Lpp-$\Delta dotA$-infected hMDMs at 6 hpi. As shown in *Figure 3A*, the $\Delta\psi_m$ collapsed specifically and significantly in WT-infected cells (*Figure 3B and C*) compared to noninfected (p=0.0022) and $\Delta dotA$-infected cells (p=0.0238), further confirming that the $F_O$-$F_1$-ATPase works in the 'reverse mode' during WT infection. Indeed, the addition of BTB to Lpp-WT-infected hMDMs led to a significant reduction of the $\Delta\psi_m$ (p<0.0001) at every time point post-infection (1–10 hpi) and at the single-cell level compared to nontreated Lpp-WT-infected cells (*Figure 3D*), further confirming that conservation of $\Delta\psi_m$ during *L. pneumophila* infection is caused by induction of $F_O$-$F_1$-ATPase 'reverse mode.'

As OXPHOS cessation and $\Delta\psi_m$ collapse can trigger cell death, we reasoned that induction of the 'reverse mode' of mitochondrial ATPase by *L. pneumophila* to maintain $\Delta\psi_m$ in the absence of OXPHOS might delay cell death of infected cells. To test this hypothesis, we first measured the percentage of living cells after an FCCP treatment of 18 hr. An FCCP-induced collapse of the $\Delta\psi_m$ reduced the percentage of living cells by 32% compared to nontreated hMDMs (p=0.0094, *Figure 4—figure supplement 1A*). Then, we infected hMDMs with Lpp-WT and treated them with BTB or left them untreated, and then measured the percentage of living cells among infected cells (*Figure 4A*). Our results showed that the percentage of living, infected cells significantly decreased after 10 hpi in BTB-treated infected hMDMs compared to nontreated cells. As this reduction in the percentage of living, infected cells upon 'reverse mode' inhibition might be caused by increased cell death, we used our high-content assay to measure Annexin-V, a marker of early apoptosis, in a high number of living hMDMs during infection (*Figure 4B and C*). While addition of BTB for 24 hr did not increase the percentage of Annexin-V$^+$ cells in noninfected cells or in Lpp-$\Delta dotA$-infected cells (*Figure 4B*, *Figure 4—figure supplement 1B*), the addition of this 'reverse mode' inhibitor to Lpp-WT-infected hMDMs significantly increased the percentage of Annexin-V$^+$ cells compared to nontreated cells (p=0.0312). Addition of BTB to Lpp-$\Delta spl$-infected cells significantly increased the percentage of Annexin-V$^+$ cells compared to nontreated cells (p=0.0375, *Figure 4C*), but no significant difference was observed compared to WT-infected cells (p=0.0571, *Figure 4—figure supplement 1B*), suggesting that *Lp*Spl is not the only effector playing a role. Thus, inhibition of the 'reverse mode' by BTB leads to a reduction in the percentage of living, infected cells, as increased cell death

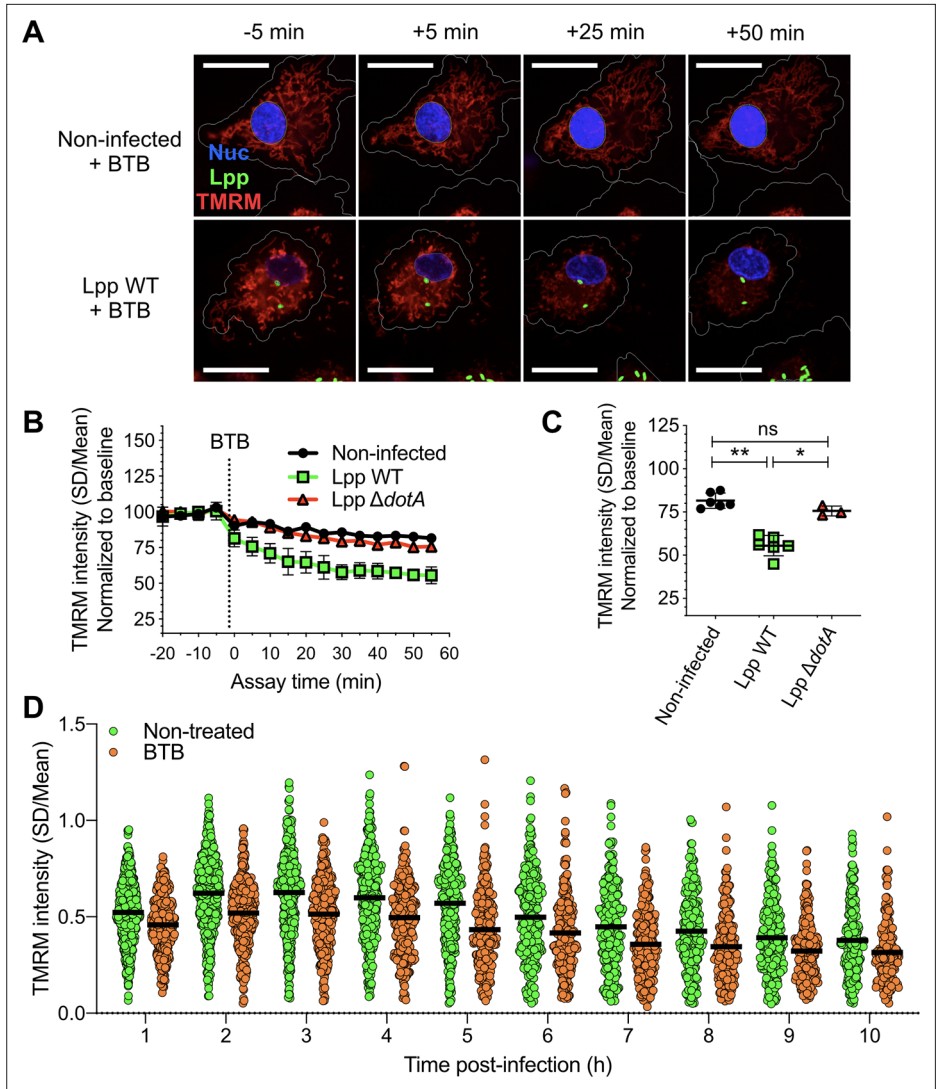

**Figure 3.** Inhibition of the 'reverse mode' of mitochondrial $F_0F_1$ ATPase reduces the $\Delta\psi_m$ of *L. pneumophila*-infected human monocyte-derived macrophages (hMDMs). (**A**) hMDMs were infected with GFP-expressing bacteria (green), Lpp-WT or Lpp-$\Delta dotA$, or left uninfected (noninfected). At 5.5 hr post-infection (hpi), cells were labeled with Hoechst to identify cell nucleus (Nuc, blue) and tetramethylrhodamine methyl ester (TMRM) (red) to quantify $\Delta\psi_m$. At 6 hpi, BTB06584 (BTB, 50 µM), a specific inhibitor of the 'reverse mode' of the ATPase, was added and $\Delta\psi_m$ monitored. Representative confocal microscopy images of noninfected (top) and Lpp-WT-infected (bottom) hMDMs (6 hpi) at 5 min before the addition and at 5, 25, and 50 min after the addition of BTB. Bar: 20 µm. (**B**) Quantification of (**C**) before (baseline) and after the addition of BTB. Each dot represents the mean ± SD of three independent experiments with a total of six replicates. (**C**) Same infection conditions than (**A**) but TMRM values (SD/mean) at 50 min after BTB addition are shown. Data from three experiments with a total of six replicates (three replicates for Lpp-$\Delta dotA$). (**D**) Single-cell analysis of $\Delta\psi_m$ in Lpp-WT-infected hMDMs treated with BTB (50 µM) or left untreated (nontreated). Single-cell data from one representative experiment *p<0.05; **p<0.01; ns, nonsignificant (Mann–Whitney *U* test).

occurs specifically in infected cells. Single-cell analysis at 12 and 18 hpi (*Figure 4D*, *Figure 4—figure supplement 1C*) also showed higher levels of Annexin-V intensity in BTB-treated Lpp-WT-infected hMDMs compared to nontreated infected cells (p<0.0001). BTB-treated infected cells also showed higher Hoechst nuclear levels compared to nontreated infected cells, a sign of nuclear condensation typical of apoptotic cells (*Figure 4—figure supplement 1D*), which further indicates that inhibition of the *Legionella*-induced ATPase 'reverse mode' ignites cell death of infected macrophages. Taken

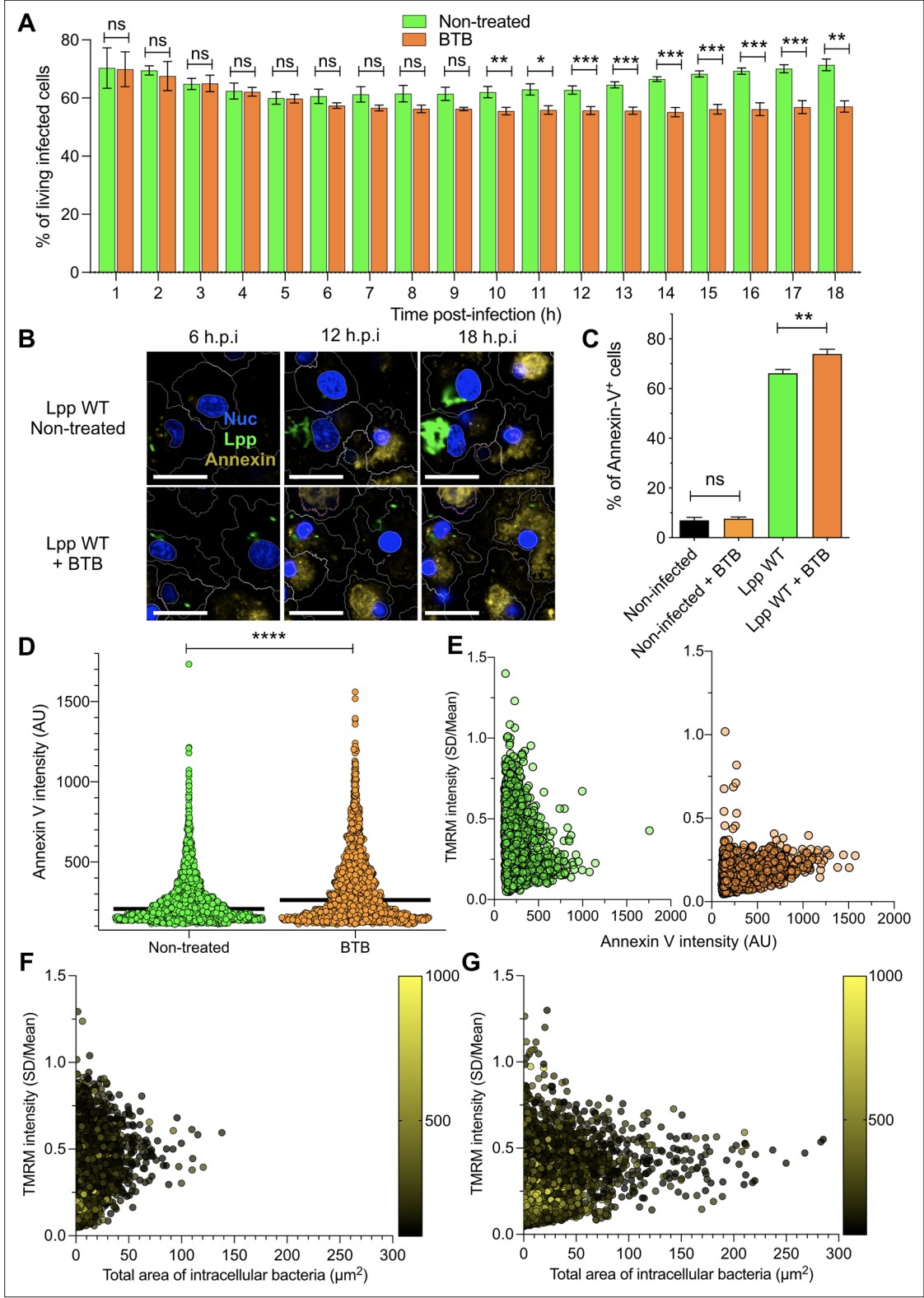

**Figure 4.** Inhibition of $F_O$-$F_1$ ATPase 'reverse mode' increases cell death in *L. pneumophila*-infected human monocyte-derived macrophages (hMDMs). (**A**) hMDMs were infected with Lpp-WT-GFP and were nontreated or treated with 50 µM BTB. The presence of GFP-expressing bacteria in each cell was monitored, and the number of living infected cells in the whole population was graphed as a percentage of living infected cells. Data from three independent experiments with a total of seven replicates per condition and time point. (**B**) hMDMs were infected with Lpp-WT-GFP (green), the nuclei of host cells were stained with Hoechst (Nuc, blue), and Annexin-V Alexa Fluor 647 was added to the cell culture to monitor early cell death (Annexin,

*Figure 4 continued on next page*

*Figure 4 continued*

yellow) from 1 to 18 hr post-infection (hpi) in nontreated or BTB-treated hMDMs. Representative confocal images of nontreated and Lpp-WT-GFP-infected cells at 6, 12, and 18 hpi are shown. Intracellular bacterial replication can be observed in nontreated Lpp-WT-infected hMDMs at 12 and 18 hpi. Bar: 20 µm. (**C**) hMDMs stained as in (**B**) were infected with Lpp-WT-GFP or left uninfected (noninfected), and then were treated or not with BTB (50 µM). Percentage of Annexin-V$^+$ cells at 24 hpi is shown. Data from three independent experiments with a total of seven replicates per condition (**D**) Single-cell analysis (12 hpi) of Annexin-V intensity of the assays described in (**B**). Single-cell data from one representative experiment (18 hpi shown in ***Figure 2—figure supplement 1A***). (**E**) hMDMs were infected with Lpp-WT-GFP, nuclei of host cells were stained with Hoechst, and tetramethylrhodamine methyl ester (TMRM) and Annexin-V Alexa Fluor 647 were added to the cells to simultaneously monitor (1–18 hpi) $\Delta\psi_m$ and early cell death, respectively, in nontreated or BTB-treated hMDMs (representative multifield confocal images in ***Figure 2—figure supplement 1C***). Single-cell analyses (12 hpi) of $\Delta\psi_m$ (TMRM SD/mean) and cell death (Annexin-V intensity) in more than 1600 cells per condition are shown. Single-cell data from one representative experiment. Green dots, nontreated Lpp-WT-infected single cells; orange dots, BTB-treated Lpp-WT-infected single cells. (**F**) Same infection conditions as in (**E**) but using nontreated cells. Bacterial replication was monitored in each single cell infected with Lpp-WT. Single-cell analyses (12 hpi) of $\Delta\psi_m$ (TMRM SD/mean), area of intracellular GFP-expressing bacteria (µm$^2$), a proxy for intracellular replication, and cell death (Annexin-V intensity) in more than 3800 cells are shown. Single-cell data from one representative experiment; color scale (yellow) represents Annexin-V intensity per cell (AU). (**G**) Same as in (**F**) at 18 hpi. *p<0.05; **p<0.01; ***p<0.001; ****p<0.00001; ns, nonsignificant (Mann–Whitney *U* test).

The online version of this article includes the following figure supplement(s) for figure 4:

**Figure supplement 1.** Inhibition of F$_O$-F$_1$ ATPase 'reverse mode' delays cell death in *L. pneumophila*-infected human monocyte-derived macrophages (hMDMs), while transfection of *Lp*Spl into HEK-293 cells protected transfected cells from Staurosporine (STS)-induced cell death.

**Figure supplement 2.** BTB treatment, $\Delta\psi_m$, and cell death of *L. pneumophila*-infected human monocyte-derived macrophages (hMDMs).

together, our results suggest that the *Legionella*-induced 'reverse mode' of the mitochondrial F$_O$F$_1$-ATPase aids to conserve $\Delta\psi_m$ during infection to delay cell death of infected macrophages.

If the induction of the 'reverse mode' in the mitochondrial F$_O$F$_1$-ATPase prevents drops in $\Delta\psi_m$, and this delays cell death, transfection of *Lp*Spl might protect cells from exogenously induced cell death. To test this hypothesis, we transfected HEK-293 cells with a plasmid expressing *Lp*Spl or with a plasmid expressing a catalytically inactive *Lp*Spl protein (*K366A) and analyzed their resistance to Staurosporine (STS)-induced cell death (***Figure 4—figure supplement 1E***). Our analyses showed that *Lp*Spl-WT-transfected HEK-293 cells were significantly more resistant to STS-induced cell death compared to control plasmid-transfected cells (p=0.0008), while cells transfected with catalytic inactive *Lp*Spl-*K366A showed an intermediate phenotype but significantly different from *Lp*Spl-WT- and control plasmid-transfected cells (p=0.0269 and p=0.0380, respectively). This suggests that although the K366 residue is important, it might not be the only residue in the protein involved in the enzymatic activity of *Lp*Spl (***Figure 4—figure supplement 1F and G***). Thus, *Lp*Spl induced 'reverse mode' of the mitochondrial F$_O$F$_1$-ATPase might protect cells from infection-independent forms of cell death.

To understand at the single-cell level whether the *Legionella*-induced 'reverse mode' of the mitochondrial F$_O$F$_1$-ATPase aids to conserve the $\Delta\psi_m$ during infection to delay cell death of infected macrophages, we simultaneously monitored $\Delta\psi_m$ and early cell death signs in the same infected cell by multiplexing Annexin-V and TMRM signals in our living cell assay (***Figure 4E***, ***Figure 4—figure supplement 2A and B***). *L. pneumophila*-infected macrophages where the 'reverse mode' activity of the F$_O$F$_1$-ATPase was inhibited by BTB suffered a collapsed $\Delta\psi_m$ and showed higher levels of Annexin-V at 12 and 18 hpi (***Figure 4E***, ***Figure 4—figure supplement 2B***) compared to nontreated cells. Thus, both events, collapse of the $\Delta\psi_m$ and triggered cell death, occurred in the same infected cell when the *Legionella*-induced 'reverse mode' activity of the F$_O$F$_1$-ATPase was inhibited. Furthermore, we correlated the area of GFP-expressing bacteria, a proxy for intracellular replication, with the $\Delta\psi m$ and cell death at the single-cell level (12 and 18 hpi, ***Figure 4F and G***, respectively). Indeed, an increased area of GFP fluorescence was observed at 18 hpi mainly in those infected macrophages with intermediate levels of TMRM (noncollapsed $\Delta\psi m$) and low Annexin-V levels (yellow color scale, ***Figure 4F and G***). This indicated that conservation of $\Delta\psi m$ and delayed cell death are required for bacterial replication as they prolong macrophage survival.

## Discussion

We show here that by inducing the 'reverse mode' of the mitochondrial F$_O$F$_1$-ATPase *L. pneumophila* circumvents the collapse of $\Delta\psi_m$ and cell death caused by OXPHOS cessation in infected cells. This mechanism, which partially involves the T4SS effector *Lp*Spl, maintains the $\Delta\psi_m$ and delays host cell death during infection, thus preserving bacterial replication niches in conditions where mitochondrial

respiration is abruptly reduced. Indeed, not only *L. pneumophila*, but also several other intracellular bacterial pathogens, such as *M. tuberculosis* or *Chlamydia pneumoniae*, reduce mitochondrial OXPHOS during infection (*Siegl et al., 2014*; *Singh et al., 2012*; *Escoll and Buchrieser, 2019*). OXPHOS reduction allows the pathogen to redirect cellular resources from mitochondria to the cytoplasm, which enhances glycolysis and biosynthetic pathways that can provide intracellular bacteria with resources needed for bacterial growth (*Escoll and Buchrieser, 2018*; *Russell et al., 2019*). In contrast, OXPHOS cessation in macrophages also enhances the biosynthetic pathways, leading to the synthesis of cytokines and antimicrobial compounds (*O'Neill and Pearce, 2016*; *Russell et al., 2019*). Furthermore, OXPHOS reduction may trigger profound consequences for host cells, such as the collapse of $\Delta\psi_m$ that may lead to subsequent cell death of infected cells. For macrophages, cell death is considered as a defense mechanism against infection (*Chow et al., 2016*). Indeed, pyroptosis of infected macrophages permits the spread of inflammatory mediators such as IL-1β. Thus, for intracellular bacteria, many of which infect macrophages (*Mitchell et al., 2016*), the death of their host cell is an obstacle as their cellular replication niche is destroyed. Therefore, while bacterial-induced reduction of OXPHOS might be beneficial for intracellular bacteria to obtain host cell resources, they need to counterbalance the consequences of OXPHOS cessation, that is, the collapse of the mitochondrial $\Delta\psi_m$ and subsequent cell death, to preserve their replication niches.

We have previously shown that the *L. pneumophila* T4SS effector MitF is implicated in fragmenting the mitochondrial networks of infected macrophages. These changes in the mitochondrial dynamics have a profound impact on OXPHOS that was severely impaired and accompanied by increased glycolysis in *Legionella*-infected cells (*Escoll et al., 2017b*). Here, we show that, despite the impairment of mitochondrial respiration in infected cells, *L. pneumophila* conserves the $\Delta\psi_m$ of host cells by inducing the 'reverse mode' of the $F_OF_1$-ATPase by a mechanism that is T4SS-dependent and partially mediated by the T4SS effector *Lp*Spl. When translocated into human cells, the S1P-lyase activity of *L. pneumophila Lp*Spl reduces S1P levels in infected cells and restrains autophagy, likely because S1P is involved in the initiation of autophagosome formation at MAMs (*Rolando et al., 2016*).

How *Lp*Spl may regulate the activity of the $F_OF_1$-ATPase is an interesting question. Our previous analyses of *Lp*Spl function have shown that this protein encodes S1P lyase activity and alters the lipid profile of the host cell by decreasing sphingosine levels in infected THP-1 cells (*Rolando et al., 2016*). Indeed, phosphorylated lipids are critical regulators of mitochondrial functions and S1P is a potent lipid mediator that regulates various physiological processes as well as diverse mitochondrial functions such as mitochondrial respiration, ETC functioning, or mitochondrial-dependent cell death (*Hernández-Corbacho et al., 2017*; *Nielson and Rutter, 2018*). Furthermore, it was reported that S1P interaction with Prohibitin 2 (PHB2) regulates ETC functioning and mitochondrial respiration (*Strub et al., 2011*) and that a link between PHB2, ETC functioning and the activation of 'mitoflashes' (*Jian et al., 2017*) exists, which are dynamic and transient uncouplings of mitochondrial respiration from ATP production that are partially dependent on the 'reverse mode' of the $F_OF_1$-ATPase (*Wei-LaPierre and Dirksen, 2019*). Thus, it is possible that *Lp*Spl modulates mitochondrial S1P levels helping the induction of the 'reverse mode' of the mitochondrial $F_OF_1$-ATPase by involving PHB2, ETC complex assembly, or the generation of mitoflashes, a fascinating possibility that we will further investigate.

The regulation of host cell death by intracellular bacteria is widely studied (*Rudel et al., 2010*). For *L. pneumophila*, T4SS effectors activating and inhibiting cell death of infected cells have been described (*Speir et al., 2014*), suggesting that a very delicate interplay of positive and negative signals governs the fate of infected macrophages. Here, we have shown that bacterial replication occurs preferentially in those infected macrophages that are able to conserve the $\Delta\psi_m$ and delay cell death, a condition that is difficult to achieve in the absence of mitochondrial respiration. Thus, the manipulation of the activity of the mitochondrial $F_OF_1$-ATPase by *L. pneumophila*, which allows this pathogen to use the ATP hydrolase activity to pump $H^+$ to the IMS to maintain the $\Delta\psi_m$ in infected cells, is a novel virulence strategy that might contribute to the fine-tuning of the timing of host cell death during bacterial infection.

## Materials and methods

### Human primary cell cultures

Human blood was collected from healthy volunteers under the ethical rules established by the French National Blood Service (EFS). Peripheral blood mononuclear cells (PBMCs) were isolated by Ficoll-Hypaque density-gradient separation (Lympholyte-H; Cedarlane Laboratories) at room temperature. PBMCs were incubated with anti-human CD14 antibodies coupled to magnetic beads (Miltenyi Biotec) and subjected to magnetic separation using LS columns (Miltenyi Biotec). Positive selected CD14$^+$ cells were counted, and CD14 expression was analyzed by flow cytometry, repeatedly showing a purity >90%. CD14 cells were plated in RPMI 1640 medium (Life Technologies) supplemented with 10% heat-inactivated fetal bovine serum (FBS, Biowest) in six-well multi-dish Nunc UpCell Surface cell culture plates or 10 cm Nunc UpCell Surface cell culture dishes (Thermo Fisher) and differentiated to hMDMs by incubation with 100 ng/ml of recombinant human macrophage colony-stimulating factor (rhMCSF, Miltenyi Biotec) for 6 days at 37°C with 5% $CO_2$ in a humidified atmosphere. At day 3, additional rhMCSF (50 ng/ml) was added. After 6 days differentiation, UpCell plates were placed at 20°C during 10 min and hMDMs were gently detached, counted, and plated in RPMI 1640 10% FBS in 384-well plates (Greiner Bio-One).

### Bacterial strains, mutant construction, and complementation

*L. pneumophila* strain Paris or JR32 and their derivatives were grown for 3 days on N-(2-acetamido)-2-amino-ethanesulfonic acid (ACES)-buffered charcoal-yeast (BCYE) extract agar, at 37°C. For eGFP-expressing strains harboring pNT28 plasmid (*Tiaden et al., 2007*), chloramphenicol (Cam; 5 µg/ml) was added. Knock-out mutant strains of *L. pneumophila* genes coding for the T4SS effectors *Lp*Spl and MitF/LegG1 were previously described (*Escoll et al., 2017b*; *Rolando et al., 2016*; *Rothmeier et al., 2013*). Strains used for complementation experiments (Lpp-WT-expressing empty pBCKS vector, Lpp-Δ*spl*-expressing pBCKS vector, and Lpp-Δ*spl*-expressing pBCKS-*spl* vector, i.e., complemented strain) were previously described (*Rolando et al., 2016*). The knock-out mutant strain of the *L. pneumophila* gene coding for the effector LncP was constructed as previously described (*Brüggemann et al., 2006*; *Rolando et al., 2016*). In brief, the gene of interest was inactivated by introduction of an apramycine resistance (apraR) cassette into the chromosomal gene by three-step PCR. The g primers used for the *lncP* (lpp2981) knock-out mutant are as follows: LncP_F: ACCCTGGTTCAT GGTAACAATGG; LncP_Inv_R: GAGCGGATCGGGGATTGTCTTATCAGGCGAATGGTGTGAAAGG; LncP_Inv_F: GCTGATGGAGCTGCACATGAAACGTCATGGTCGTGCTGGTTG; LncP_R: AATCAGAT GGGTAAGCCGATTGG. To amplify the apramycine cassette, the primers Apra_F: TTCATGTGCAGC TCCATCAGC and Apra_R: AAGACAATCCCCGATCCGCTC were used.

### Infection of hMDMs

hMDMs were infected with *L. pneumophila* grown for 3 days on BCYE agar plates. Bacteria were dissolved in 1× PBS (Life Technologies), the optical density (OD) was adjusted to $OD_{600}$ of 2.5 (2.2 × 10$^9$ bacteria/ml), and the bacteria were then further diluted in serum-free XVIVO-15 medium (Lonza) prior to infection to obtain the respective multiplicity of infection (MOI). hMDMs were washed twice with serum-free XVIVO-15 medium and then infected (MOI = 10) with 25 µl of bacteria in 384-well plates (Greiner Bio-One). The infection was synchronized by centrifugation (200 × *g* for 5 min), and the infected cells were incubated at 37°C for 5 min in a water bath and then for 25 min at 37°C/5% $CO_2$. After three intensive washes with serum-free XVIVO-15 medium, the infection proceeded in serum-free XVIVO-15 medium for the respective time points.

### Infection and transfection of HEK-293 cells

HEK-293 cells stably expressing the FcγRII receptor (gift of Prof. Craig Roy) (*Arasaki and Roy, 2010*) were maintained in DMEM +10% FBS. Before use, the cells were tested negative for *Mycoplasma* contamination. The HEK-293 cells stably expressing the FcγRII receptor were not authenticated. However, the only entry of misidentified HEK cells at the ICLAC database is ICLAC-00063, when HEK cells were misidentified in 1981 with HeLa cells. Therefore, our HEK cells were identified by morphology and attachment as HEK cells are morphologically different from HeLa cells and they attach very weakly to tissue culture-treated plastic, contrary to HeLa cells. Most importantly, the remote possibility of misidentification with HeLa cells would not change any consideration or conclusion in our study. For

infection, HEK-293 cells were plated in 384-well plates (Greiner Bio-One) and were infected with *L. pneumophila* grown for 3 days on BCYE agar plates following the same protocol used for hMDMs but using MOI = 20 without washing to avoid detaching of cells. The infection proceeded in serum-free XVIVO-15 medium for the respective time points. For transfection, HEK-293 cells were plated in 384-well plates (Greiner Bio-One) and FuGENE (Promega) and Opti MEM medium (Thermo Fisher) were used, following the manufacturer's recommendations. We used 1 µg DNA + 3 µl FuGENE + 500 µl Opti MEM, and then added 2.5 µl per well. Plasmid DNA was transfected during 24 hr. Transfected plasmids expressed *Lp*Spl WT (harboring an Xpress tag) or a catalytically inactive *Lp*Spl protein (*K366A, also harboring an Xpress tag), previously described (*Rolando et al., 2016*). The pGFPmax plasmid (Lonza) was used as a control (as cells transfected with a plasmid only expressing the small epitope Xpress, without any other protein, gave extremely low-intensity values during immunofluorescence experiments and could not be used as a control).

## Automatic confocal imaging

Cell imaging was performed in 384-well plates. For living cells, 30 min prior imaging, 25 µl of culture medium were removed and replaced by 25 µl of 2× mix of dyes, to a final concentration of 200 ng/ml of Hoechst H33342 (nuclear staining; Life Technologies), 10 nM of TMRM (mitochondrial membrane potential; Life Technologies), and/or 1/100 Annexin-V-Alexa Fluor 647 (early apoptosis; Life Technologies). If chemical inhibitors of the ETC were used in the experiments, they were added to hMDMs at the indicated times points at the following concentrations: 5 µM oligomycin (Enzo), 100 µM DCCD (Sigma), 10 µM FCCP (Tocris), and 50 µM BTB06584 (Sigma). Once the assay was performed, cells were fixed with 4% PFA, permeabilized with 0.1% Triton-X100, blocked with 1% BSA, and stained with primary mouse antibodies against Xpress tag (1:100, Invitrogen) and secondary anti-mouse Alexa Fluor 488 antibodies (1:1000, Invitrogen). Image acquisitions of multiple fields (9–25) per well were performed on an automated confocal microscope (Opera Phenix, PerkinElmer) using ×40 or ×60 objective, excitation lasers at 405, 488, 561, and 640 nm, and emission filters at 450, 540, 600, and 690 nm, respectively.

## Metabolic extracellular flux analysis

hMDMs (50,000) were plated in XF-96-cell culture plates (Seahorse Bioscience). For OCR measurements, XF Assay Medium (Seahorse Bioscience) supplemented with 1 mM pyruvate and 10 mM glucose was used, and OCR was measured in a XF-96 Flux Analyzer (Seahorse Bioscience). For the mitochondrial respiratory control assay, hMDMs were infected at MOI = 10 and at 6 hpi. Different drugs were injected (Mitostress kit, Seahorse Bioscience) while OCR was monitored. Specifically, olygomycin was injected through port A, then FCCP was injected through port B, and finally rotenone + antimycin A were injected through port C, to reach each of the drugs a final concentration in the well of 0.5 µM. Coupling efficiency (%) was calculated using the following formula: (ATP production rate)/(basal respiration rate) × 100. In this formula, ATP production rate was calculated as: (last rate measurement before oligomycin injection) – (minimum rate measurement after oligomycin injection), while basal respiration rate was calculated as: (last rate measurement before first injection) – (nonmitochondrial respiration rate).

## Automatic high-content analyses (HCA)

All analyses were performed with Harmony software v.4.9 (PerkinElmer) using in-house scripts developed in Harmony running Acapella Assay Language version 5.0.1.124082 (available at https://github.com/bbi-ip/Legionella_and_mitochondrial_ATPase.git; *Rusniok, 2021*; copy archived at swh:1:rev:657b662b912e3b3c630619f648489cc28652dcd9) . For the HCA of the mitochondrial membrane potential ($\Delta\psi_m$), the Hoechst signal was used to segment nuclei in the 405/450 channel (excitation/emission), Hoechst background signal in the cytoplasm was used to segment the cytoplasm region in the 405/450 channel, *L. pneumophila* was identified by measuring the GFP signal in the 488/540 channel, and TMRM (10 nM) signal in the 561/600 channel was used to measure $\Delta\psi_m$ by calculating SD/mean TMRM intensity values in each infected and noninfected cell. For the HCA of cell death, the Hoechst signal was used to segment nuclei in the 405/450 channel, Hoechst background signal in the cytoplasm was used to segment the cytoplasm region, and the identification of *L. pneumophila* was performed using the GFP signal in the 488/540 channel. Then, Annexin-V-Alexa Fluor

647 signal was measured in the 640/690 channel and the Hoechst signal intensity was measured in the 405/450 channel for each infected or noninfected cell. For the HCA combining $\Delta\psi_m$ and cell death, both the aforementioned HCA strategies were merged using high Hoechst signal in the 405/450 channel to segment nuclei, low Hoechst signal in the 405/450 channel to segment cytoplasm, GFP signal in the 488/540 channel to identify bacteria, TMRM signal in the 561/600 channel to measure $\Delta\psi_m$ (SD/mean), and Annexin-V-Alexa Fluor 647 signal in the 640/690 channel to measure cell death. For the HCA of transfected cells, GFP/Xpress-488 signal in the 488/540 channel was used to identify transfected cells and Annexin-V-Alexa Fluor 647 signal in the 640/690 channel to measure cell death.

## Whole-genome sequencing for mutant validation

Chromosomal DNA was extracted from BCYE-grown *L. pneumophila* using the DNeasy Blood and Tissue Kit (QIAGEN). The Illumina NGS libraries were prepared using the Nextera DNA Flex Library Prep following the manufacturer's instructions (Illumina Inc). High-throughput sequencing was performed with a MiSeq Illumina sequencer (2 × 300 bp, Illumina Inc) by the Biomics Pole (Institut Pasteur). For the analysis, we first removed adapters from Illumina sequencing reads using Cutadapt software version 1.15 (**Martin, 2011**) and then used Sickle (https://github.com/najoshi/sickle, **Buffalo, 2021**) with a quality threshold of 20 (Phred score) to trim bad quality extremities. Reads were assembled using Spades (**Nurk et al., 2013**) and different K-mer values. The region corresponding to the gene of interest was identified by blastn, extracted, and compared to the homologous region in the *L. pneumophila* strain Paris WT genome and to the antibiotic cassette sequence using blastn. The results were visually inspected with Artemis Comparison Tool (ATC) (**Carver et al., 2005**). In addition, we searched the entire genome whether off-target mutations had occurred using Bowtie 2 (**Langmead and Salzberg, 2012**) to perform a mapping against the genome sequence of *L. pneumophila* strain Paris (NC_006368.1). SNPs and small indels were searched for with freebayes SNP caller (**Garrison and Marth, 2012**), and mutations and small indels were visualized in the Artemis genome viewer (**Carver et al., 2005**) to analyze them (new amino acid, synonymous mutation, frameshifts, etc.). We used Samtools to find regions with no coverage (or close to zero) (**Li et al., 2009**). Regions or positions with such anomalies were visualized and compared with the corresponding region of the assembly. This confirmed that no off-target mutations impacting the phenotype of the mutant had occurred.

## Statistical analyses

The two-sample Student's *t*-test (Mann–Whitney *U* test, nonassumption of Gaussian distributions) was used in all data sets unless stated otherwise. Data analysis was performed using Prism v9 (GraphPad Software).

## Acknowledgements

We acknowledge CB's, P Glaser's lab members, and F Stavru at Institut Pasteur for fruitful discussions. We specially thank Daniel Schator for help and discussions regarding HEK-293 infection and transfection. We thank N Aulner, A Danckaert, Photonic BioImaging (PBI) UTechS and M Hassan, Center for Translational Science (CRT) at Institut Pasteur, for support. This research was funded by the Institut Pasteur, DARRI-Institut Carnot-*Microbe et santé* (grant number INNOV-SP10-19) to PE; the Agence National de Recherche (grant number ANR-10-LABX-62-IBEID to CB and ANR-21-CE15-0038-01 to PE), the Fondation de la Recherché Médicale (FRM) (grant number EQU201903007847) to CB, and the Région Ile-de-France (program DIM1Health) to PBI (part of FranceBioImaging, ANR-10-INSB-04–01). MD was supported by the Ecole Doctorale FIRE – 'Programme Bettencourt.' SS was supported by the Pasteur Paris-University (PPU) International PhD Program.

## Additional information

### Funding

| Funder | Grant reference number | Author |
|---|---|---|
| Agence Nationale de la Recherche | ANR-10-LABX-62-IBEID | Carmen Buchrieser |
| Fondation pour la Recherche Médicale | EQU201903007847 | Carmen Buchrieser |
| Association Instituts Carnot | INNOV-SP10-19 | Pedro Escoll |
| French National Research Agency | ANR- 21- CE15- 0038- 01 | Pedro Escoll |

The funders had no role in study design, data collection and interpretation, or the decision to submit the work for publication.

### Author contributions

Pedro Escoll, Conceptualization, Formal analysis, Investigation, Supervision, Writing - original draft; Lucien Platon, Formal analysis, Investigation, Methodology; Mariatou Dramé, Silke Schmidt, Formal analysis, Investigation; Tobias Sahr, Investigation, Methodology; Christophe Rusniok, Data curation, Formal analysis; Carmen Buchrieser, Funding acquisition, Project administration, Supervision, Writing - review and editing

### Author ORCIDs

Pedro Escoll  http://orcid.org/0000-0002-5933-094X
Lucien Platon  http://orcid.org/0000-0001-7894-5977
Carmen Buchrieser  http://orcid.org/0000-0003-3477-9190

### Decision letter and Author response

Decision letter https://doi.org/10.7554/eLife.71978.sa1
Author response https://doi.org/10.7554/eLife.71978.sa2

## Additional files

### Supplementary files
• Transparent reporting form

### Data availability

All data generated or analysed during this study are included in the manuscript. Source data files are uploaded to Github: https://github.com/bbi-ip/Legionella_and_mitochondrial_ATPase.git (copy archived at https://archive.softwareheritage.org/swh:1:rev:657b662b912e3b3c630619f648489cc28652dcd9).

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
