## [Editor Report]

The pathogenic bacterium *Legionella pneumophila* (Lp) is known for its ability to translocate cocktails of effector proteins into its eukaryotic host. Yet, despite an overall reduction in mitochondrial oxidative phosphorylation following *Legionella* infection, host cell mitochondria maintain their normal membrane potential (Δψ_m_). In this study, the authors show that the translocated effector protein LpSpl forces the host cell’s FOF1-ATPase to function in ‘reverse mode,’ thereby maintaining Δψ_m_, which, ultimately, supports bacterial replication by delaying host death.

---

## [Decision Letter]

[Editors' note: this paper was reviewed by Review Commons.]

**Decision letter after peer review:**

Thank you for submitting your article "Reverting the mode of action of the mitochondrial FOF1-ATPase by Legionella pneumophila preserves its replication niche" for consideration by *eLife*. Your article has been reviewed by a Reviewing Editor and Gisela Storz as the Senior Editor.

Essential revisions:

1) Please provide experiments to further strengthen the idea that the connection between LpSpl and FoF1-ATPase isn't simply coincidental.

2) Complementation analysis should be performed for the LpSpI gene.

3) Consider including a LppΔdotA mutant control in the LppWT vs LppWT + BTB experiments, as the differences for the WT appeared subtle.

4) Related to figure 1D and E: these experiments would be strengthened by adding an additional condition whereby mitochondrial fragmentation is induced in non-infected cells (e.g., to demonstrate the efficiency of the TMRM approach in monitoring loss-of-potential).

---

## [Author Response]

Essential revisions:1) Please provide experiments to further strengthen the idea that the connection between LpSpl and FoF1-ATPase isn't simply coincidental.2) Complementation analysis should be performed for the LpSpI gene.3) Consider including a LppΔdotA mutant control in the LppWT vs LppWT + BTB experiments, as the differences for the WT appeared subtle.4) Related to figure 1D and E: these experiments would be strengthened by adding an additional condition whereby mitochondrial fragmentation is induced in non-infected cells (e.g., to demonstrate the efficiency of the TMRM approach in monitoring loss-of-potential).

For this revision we have undertaken all experiments and analyses that the reviewers had asked for and that we had suggested in our first submission. They are now added to the manuscript.

These experiments include:

– Analyses of the complemented Δ*spl* mutant strain in all experiments where this mutant was used

– Set up of HEK-293 cells as additional model for transfection as it is not possible to transfect hMDMs. We obtained the same results with respect to the F_O_-F_1_ATPase as in hMDMs.

– Analyses of the resistance of *Lp*Spl-transfected cells to Staurosporine (STS)induced cell death

– Analyses of cellular respiration of the *Δsp*l mutant strain

– Annexin assay on cells infected with the *Δsp*l (-/+ BTB) and the Δ*dotA* mutant

These additional experiments further confirm and substantiate our finding that in infection *L. pneumophila* reverses the ATP-synthase activity of the mitochondrial F_O_F_1_-ATPase to ATP-hydrolase activity in a T4SS-dependent manner as well as the implication of theT4SS effector *Lp*Spl.

[Editors' note: we include below the reviews that the authors received from Review Commons, along with the authors’ responses.]

Reviewer #1 (Evidence, reproducibility and clarity):Escoll and colleagues present an elegant study investigating mitochondrial biology in the context of cell infection with displaying Legionella pneumophila (Lp). In spite of an overall reduction in mitochondrial oxidative phosphorylation following Legionella infection, host cell mitochondria interestingly maintain a normal mitochondrial membrane potential (Δψm). The authors propose the effector protein, LpSpl is involved in the maintenance of Δψm by promoting "reverse mode" functioning of the FoF1-ATPase. It is suggested the maintenance of the membrane potential delays host death, ultimately supporting bacterial replication.The theme of this paper is of high relevance and underscores exciting cell biology, involving aspects of host-pathogen interactions and mitochondrial biology. Over all the data is convincing, well controlled and utilised some nice methodologies. The key conclusion however needs supporting data.Major comments:[1] The connection between LpSpl and FoF1-ATPase could simply be a coincidence. The authors suggest some viable options in the discussion, but it is hard to rationalise how this paper could be published in mid-range journal without exploring this functional aspect of LpSpl in more detail. Given the exogenously expressed LpSpl is localised to mitochondria, could some analysis be performed with this protein to ascertain the impact at mitochondria. For example, are cells expressing LpSPL more resistant to death? Does their lipid profile changed? Does the protein interact with FoF1-ATPase? This would be very challenging with the native bacterially-translocated protein (due to protein levels), but making use of the exogenous protein could greatly assist the paper.

To further validate the connection between LpSpl and the “reverse mode” of the mitochondrial ATPase, and in agreement with a similar comment from Reviewer #2, we have performed several of the experiments suggested by the reviewer.

First, we analyzed a complemented Δ*spl* mutant strain. In Figure 2H the results of the measurement of the forward/reverse mode using DCCD including the Δ*spl* mutant are shown. If the “reverse mode” of the F_O_F_1_-ATPase is partially due to *Lp*Spl the decreased effect observed during infection with the Δ*spl* mutant should be recovered to the WT phenotype in the complemented strain (*Lp*Spl is expressed on a plasmid). Indeed, our experiments, performed in hMDMs isolated from 3 different donors, confirm that the complemented strain reverses the effect observed with the mutant strain (new Figure 2I).

These new results are added in Lines 233-247 it reads now:

“To confirm that LpSpl is indeed implicated in the induction of the “reverse mode” of the F_O_F_1_-ATPase we complemented the Lpp-∆spl mutant strain with a plasmid expressing LpSpl (Lpp ∆spl-spl) and infected hMDMs. […] We also found a very small but significant difference in the oxygen consumption levels exhibited by Lpp-WT- and Lpp ∆spl-infected hMDMs (p=0.0148, Figure 2—figure supplement 1A), suggesting that the role of LpSpl in the induction of the “reverse mode” to maintain Δψ_m_ during infection might partially involve the modulation of the functioning of the OXPHOS machinery.”

Secondly the reviewer suggested to analyze the lipid profile, of the Δ*spl* strain compared to the WT strain. Indeed, this is important*,* however we have already published the lipid profile of the Δ*spl* strain in our previous study (Rolando et al. Proc Natl Acad Sci USA 113(7):11901-6). We showed that the levels of sphingosine were altered in infected THP1 macrophage-like cells in a *Lp*Spl-dependent manner (see Figure 3D in Rolando et al., 2016). As *Lp*Spl encodes sphingosine-1-phosphate (S1P) lyase activity, *Lp*Spl indeed decreases sphingosine levels in the host cell. In our previous work, we have shown that sphingosine levels were decreased in Lpp-WT-infected cells compared to Lpp-Δ*spl*-infected cells, and this effect was reversed when we complemented the mutant strain with a plasmid expressing LpSpl. We have added our previous result now to the text. However, as discussed in the manuscript (Lines 381-384), alteration of S1P levels in mitochondria might alter in turn ETC functioning.

Lines 373-376 it reads now:

“How LpSpl may regulate the activity of the F_O_F_1_-ATPase is an interesting question. Our previous analyses of LpSpl function has shown that this protein indeed encodes S1P lyase activity and alters the lipid profile of the host cell by decreasing sphingosine levels in infected THP-1 cells (Rolando et al., 2016).”

Thirdly the reviewer proposed to exogenously express *Lp*Spl in human cells. As it is not possible to transfect hMDMs we have performed these experiments by transfecting *Lp*Spl in HEK-293 cells. However, for this we needed to test first if our results obtained using human primary macrophages (hMDMs) are also observed in HEK-293 cells. We thus measured the direction of the F_O_-F_1_-ATPase in these cells (forward, reverse) during infection (New Figure 2—figure supplement 1B to 1F S2B to S2F). For these experiments, we have used HEK-293 cells stably expressing the FcγRII receptor (kindly provided by Prof. Craig Roy, Traffic 11 (5): 587-600), which allow efficient internalization of IgG-opsonized *L. pneumophila* (we opsonized bacteria using an antibody targeting *Legionella* flagellin). To validate Δ*ψm* measurements in HEK-293 cells, we monitored Δψm upon FCCP addition to non-infected cells, which decreased as expected (new Figure 2—figure supplement 1B and 1C). Then we measured the direction of the F_O_-F_1_-ATPase during infection using DCCD (as in Figure 2G, hMDMs). Our results showed that the Δ*ψm* decreased upon addition of DCCD in Lpp-WT-infected HEK-293 cells, confirming that the F_O_-F_1_-ATPase of WT-infected HEK-293 cells worked in the “reverse mode” (new Figure 2—figure supplement 1D). We also obtained results equivalent to those obtained in hMDMs when using the mutant strains. The results for the Δ*ψm* upon DCCD addition in *ΔdotA*-infected cells was similar to non-infected cells (meaning in “forward mode”), while results obtained in *Δspl*-infected HEK-293 cells showed an intermediate phenotype between the WT and the *ΔdotA* mutant, but significantly different to both strains (new Figure 2—figure supplement 1E and 1F, similar to our results obtained with hMDM cells shown in Figure 2H).

These experiments are added to the text. Lines 248-262 it reads now:

“To further analyze our results obtained in hMDMs in an easy to transfect cell model we used HEK-293 cells stably expressing the FcγRII receptor (Arasaki and Roy, 2010) as these cells allow efficient internalization of opsonized L. pneumophila. […] The change of the Δψ_m_ upon DCCD addition in ΔdotA-infected cells were similar to those in non-infected cells, while results obtained in Δspl-infected HEK-293 cells showed an intermediate phenotype between the WT and the ΔdotA mutant, but significantly different to both strains (Figure 2—figure supplement 1E and 1F).”

After we had validated our results in HEK-293 cells, we performed the experiments suggested by reviewer 1 and analyzed the resistance of *Lp*Spl-transfected cells to Staurosporine (STS)-induced cell death. We transfected HEK-293 cells with a plasmid expressing *Lp*Spl harboring a small epitope peptide, the Xpress tag (used in our previous study Rolando et al. Proc Natl Acad Sci USA 113(7):11901-6), which allowed us to analyze by immunofluorescence exclusively those cells that were indeed transfected by using an anti-Xpress antibody. In parallel, we transfected a plasmid expressing a catalytic inactive form of *Lp*Spl (mutation *K366A, Rolando et al. Proc Natl Acad Sci USA 113(7):11901-6), which also expresses the Xpress tag to exclusively analyze transfected cells (new Figure 4—figure supplement 1E). As immunofluorescence of the Xpress tag in cells transfected with a plasmid that only expresses this small epitope (without any other protein) gave extremely low intensity values, we were unable to use this plasmid as a control for our analyses. We thus decided to use a plasmid expressing GFP as control (control pMax-GFP, Lonza). Our results demonstrated that *Lp*Spl-WT-transfected HEK-293 cells were more resistant to STS-induced cell death compared to control plasmid-transfected cells, while cells transfected with mutated *Lp*Spl-*K366A showed and intermediate phenotype, at the level of % of Annexin V+ cells as well as at the level of Annexin-V fluorescence intensity, suggesting that although the K366 residue is important, is not the only site involved in the function of *Lp*Spl (new Figures S3F and S3G). These new results have been added as figures S3E, F and G and in the text.

Lines 309 – 322 it reads now:

“If the induction of the “reverse mode” in the mitochondrial F_O_F_1_-ATPase prevents drops in Δψ_m_, and this delays cell death, transfection of LpSpl might protect cells from exogenously induced cell death. […] Thus, LpSpl induced “reverse mode” of the mitochondrial F_O_F_1_-ATPase might protect cells from infection-independent forms of cell death.”

Taken together, our new results shown in Figures S2 and S3 strengthen the manuscript by adding a new cell type that reproduces the model shown in hMDMs with respect to the role of *Lp*Spl in the functioning of the F_O_-F_1_-ATPase during infection and by showing the effects of transfected *Lp*Spl protein in the resistance of cells to exogenously-induced cell death. The new methods used and associated details have been added to the Methods section (Lines 444-458-546, 508-510).

Minor comments:[1] Is the phenotype presented also apparent in JR32 (lines 113 and 114). If the answer to this is known would be good to clarify in text.

Yes, the phenotype presented here is also apparent in *L. pneumophila* strain Philadelphia JR32. These results are shown in Figure 2H and described in Line 219 Now we included a call to Figure 2H in the sentence and highlighted the result with an additional sentence.

Lines 220-225 reads now:

“the F_O_F_1_-ATPase worked in the “reverse mode” during infection with the Lpp-WT and JR32-WT strains (Figure 2G and 2H).As infection of hMDMs by L. pneumophila strain Paris and L. pneumophila strain Philadelphia JR32 show similar results, this suggests that the inhibition of mitochondrial respiration and the conservation of the Δψm through the induction of the “reverse mode” of the mitochondrial ATPase is a virulence strategy of L. pneumophila.”

[2] How does the deltaSpl strain look using the cellular respiration assay (Figure 1). Showing the profile of the different strains with this assay would be useful.

We thank the reviewer for this suggestion. We have performed these experiments and have added the results as new Figure S2A (old Figure S2 is now Figure S3). As shown in new Figure S2A, cellular respiration of the Lpp-Δ*spl* mutant is similar to Lpp-WT values with a small but statistical significant difference between both (Lpp-WT vs. Lpp-Δ*spl* p-value = 0,0022). This is in line with our results that suggest that *Lp*SPL affects the hydrolase activity of the F_O_F_1_-ATPase (reverse mode), which should be downstream of OCR inhibition. In our model, OXPHOS cessation is partially mediated by MitF-dependent mitochondrial fragmentation (Escoll et al. 2017). Therefore, mutating the *spl* gene in *Legionella* may not have an effect on the OCR of infected hMDMs. In addition, as shown in Figure 2H, the effect of *Lp*SPL in the induction of the “reverse mode” is partial, suggesting that also other effectors and/or mechanisms participate, further providing an explanation why hMDMs infected with *L. pneumophila*-WT and Δ*spl* mutant strain show similar OCR values. This result is now discussed.

Lines 243-247 it reads now:

“We also found a very small but significant difference in the oxygen consumption levels exhibited by Lpp-WT- and Lpp ∆spl-infected hMDMs (p=0.0148, Figure 2—figure supplement 1A), suggesting that the role of LpSpl in the induction of the “reverse mode” to maintain Δψ_m_ during infection might partially involve the modulation of the functioning of the OXPHOS machinery.”

[4] Cell death measurements. The annexin assay should also be performed on cells infected with deltaSpl (-/+ BSB). This would strengthen the connection between LpSp1 and the overall cell phenotype.

As requested, we have performed these experiments, also adding the Lpp-Δ*dotA* mutant as Reviewer #2 suggested (new Figure S3B). Our results showed that, contrary to Lpp-WT-infected cells, inhibition of the reverse mode using BTB in Lpp-Δ*dotA*-infected cells did not significantly increase cell death. Results obtained during infection with the Lpp-Δ*spl* mutant were not significantly different from WT, although they were very close (p = 0,0571). These results might highlight the redundancy known in the *Legionella* repertoire of >300 bacterial effectors. Effector redundancy is a typical phenomenon for *L. pneumophila* infection (studied by several groups, e.g. O'Connor TJ, Boyd D, Dorer MS, Isberg RR. “Aggravating genetic interactions allow a solution to redundancy in a bacterial pathogen”. Science. 2012 Dec 14;338(6113):1440). Thus, if one single virulence factor is deleted, it is rare that one can observe strong phenotypes because of redundancy. Indeed, several other effectors have been shown to modulate host cell death, such as VipD, SidF, or Ceg18 (Speir et al. Future Medicine 2014; 9(1): 107-118). We think this might be the reason that no significant difference between Lpp-WT- and Lpp-Δ*spl*-induced cell death are seen.

These results are added Lines 291-298, it reads now:

“While addition of BTB for 24 hours did not increase the percentage of Annexin-V^+^ cells in non-infected cells or in Lpp-ΔdotA-infected cells (Figure 4B, Figure 4—figure supplement 1B), the addition of this “reverse mode” inhibitor to Lpp-WT-infected hMDMs significantly increased the percentage of Annexin-V^+^ cells compared to non-treated cells (p = 0.0312). Addition of BTB to Lpp-Δspl-infected cells significantly increased the percentage of Annexin-V^+^ cells compared to non-treated cells (p=0.0375, Figure 4C), but it was not significantly different compared to WT infected cells (p=0,0571, Figure 4—figure supplement 1B), suggesting that LpSpl is not the only effector playing a role.”

[5] Line 167-169 maybe reference some papers on the reverse action of the FoF1-ATPase for those that might not be familiar with this phenomenon.

As requested, now line 171 a reference on the “reverse mode” of the FoF1 ATPase was added (Michelangelo Campanella et al. Trends in Biochemical Sciences).

Significance:The theme of this paper is of high relevance and underscores exciting cell biology, involving aspects of host-pathogen interactions and mitochondrial biology. Over all the data is convincing, well controlled and utilised some nice methodologies. The key conclusion however needs supporting data.

Thank you for the kind words, we hope that the new results and new discussions added to the manuscript now support well the conclusions.

Referee Cross-commenting:I agree with Reviewers 2 and 3 that there is many common themes to our suggestions and likewise have no further comments.Reviewer #2 (Evidence, reproducibility and clarity):The authors firstly analyzed oxygen consumption rate (OCR) in human monocyte-derived macrophages (hMDMs) upon Legionella pneumophila infection with sequential addition of mitochondrial respiratory inhibitors, revealing that the rate of mitochondrial respiration coupled to ATP synthesis is highly reduced in infected cells. Based on the assumption that the F0F1-ATPase activity can influence the mitochondrial membrane potential (Δψm), they conducted the TMRM fluorescence-based kinetic measurement of the Δψm. The result of the analysis suggested that L. pneumophila can maintain the Δψm by manipulating the mitochondrial electron transport chain (ETC) under the basal condition despite a significant reduction of oxidative phosphorylation (OXPHOS). On these bases, they conducted the TMRM analysis combined with usage of the F0F1-ATPase inhibitors to identify which activity mode the F0F1-ATPase has ("Forward" or "Reverse") when the cells are infected with L. pneumophila. The kinetics of the Δψm was consistent with their model that L. pneumophila infection induces the "Reverse Mode" of the ATPase, which can be partly mediated by a L. pneumophila effector LpSpl. They also utilized the "Reverse Mode" specific inhibitor, BTB, to further support the model. Finally, they conducted the detection of early apoptosis by measuring Annexin-V with or without BTB-treatment. From the data, they concluded that the biological significance of the L. pneumophila-induced "Reverse Mode" of mitochondrial ATPase was for prevention of early cell death to preserve the replication niche.Major comments:(1) The involvement of LpSpl for maintaining the Δψm was supported by only one data (Figure 2H). At least a complementation analysis (by providing a plasmid expressing Spl) should be required to ensure the reliability.

As suggested by Reviewer 2 (and Reviewer 1) we have performed the requested experiments in hMDMs obtained from 3 different donors. Our results confirmed that the complemented strain reverses the effect observed with the mutant strain. These results are shown in new Figure 2I and answer to point 1 of reviewer 1.

(2) In Figure 4C, the difference between LppWT vs LppWT + BTB is subtle. I would like to see if the control set, LppΔdotA vs LppΔdotA +BTB, has no significant difference in the cell death.

As requested, we have performed these experiments, also adding the Lpp-Δ*spl* mutant as Reviewer #1 suggested (new Figure 4—figure supplement 1B). Our results showed that, contrary to Lpp-WT-infected cells, inhibition of the reverse mode using BTB in Lpp-Δ*dotA*-infected cells did not significantly increase cell death. Results obtained during infection with the Lpp-Δ*spl* mutant were not significantly different from WT, although they were very close (p = 0,0571). These results might highlight the redundancy known in the *Legionella* repertoire of >300 bacterial effectors. Effector redundancy is a typical phenomenon for *L. pneumophila* infection (studied by several groups, e.g. O'Connor TJ, Boyd D, Dorer MS, Isberg RR. “Aggravating genetic interactions allow a solution to redundancy in a bacterial pathogen”. Science. 2012 Dec 14;338(6113):1440). Thus, if one single virulence factor is deleted, it is rare that one can observe strong phenotypes because of redundancy. Indeed, several other effectors has been shown to modulate host cell death, such as VipD, SidF, or Ceg18 (Speir et al. Future Medicine 2014; 9(1): 107-118). We think this might be the cause for not a significant difference between Lpp-WT- and Lpp-Δ*spl*-induced cell death.

These results are added in Lines 291-298, it reads now:

“While addition of BTB for 24 hours did not increase the percentage of Annexin-V^+^ cells in non-infected cells or in Lpp-ΔdotA-infected cells (Figure 4B, Figure 4—figure supplement 1B), the addition of this “reverse mode” inhibitor to Lpp-WT-infected hMDMs significantly increased the percentage of Annexin-V^+^ cells compared to non-treated cells (p = 0.0312). Addition of BTB to Lpp-Δspl-infected cells significantly increased the percentage of Annexin-V^+^ cells compared to non-treated cells (p=0.0375, Figure 4C), but no significant difference was observed compared to WT infected cells (p=0,0571, Figure 4—figure supplement 1B), suggesting that LpSpl is not the only effector playing a. role.”

(3) In Figure 1D, a control showing an extensive alteration of the TMRM intensity may need to be included to support the statement " no differences between the infection conditions" (line 157).

As requested, we have performed these experiments (also asked for by Reviewer #3) in hMDMs isolated from 3 different donors. They confirmed our results (new Figure 1E and S1D). Our results in hMDMs isolated from Donor #2 using FCCP to completely depolarize mitochondria have been included in Figure 1E, and the entire single-cell dataset from the 3 donors used for these experiments has been added to the supplement as Figure 1—figure supplement 1D.

We have updated the text in Lines 159-164, it reads now:

“Single-cell analyses (Figure 1E and S1C) showed that Lpp-WT-, Lpp-∆dotA- and non-infected single hMDMs showed a wide range of Δψ_m_ values at any time-point (Figure 1—figure supplement 1C) with no significant differences between Lpp-WT- and Lpp-∆dotA-infected hMDMs at 6 hpi. Infected cells with both strains showed a significantly higher Δψ_m_ compared to non-infected cells (p<0.0001, Figure 1E, Figure 1 – —figure supplement 1D).”

Minor comments:(1) I assume that Figures 4FG were conducted without BTB. But I could not find the description in the text or in the figure legend.

Indeed, thank you for pointing out this mistake. We have corrected it now.

Line 777-781 now reads:

“(F) Same infection conditions as in (E) but using non-treated cells. Bacterial replication was monitored in each Lpp-WT-infected single cell. Single-cell analyses (12 hpi) of Δψm (TMRM SD/Mean), area of intracellular GFP-expressing bacteria (μm^2^), a proxy for intracellular replication, and cell death (Annexin-V intensity) ….”

(2) I could not understand how the "Coupling Efficiency" was calculated in Figure S1B. And what "coupling" it exactly means?

Coupling efficiency (%) was calculated using the following formula:

(ATP Production Rate) / (Basal Respiration Rate) × 100

In this formula, the ATP production rate is the magenta square of Figure 1 —figure supplement 1A and is calculated as: (Last rate measurement before Oligomycin injection) – (Minimum rate measurement after Oligomycin injection), while Basal Respiration Rate is the blue square of Figure 1—figure supplement 1A and is calculated as: (Last rate measurement before first injection) – (Non-Mitochondrial Respiration Rate). To obtain the coupling efficiency (%), these formula were applied to the data obtained in Figure 1A (and outlined in Figure 1—figure supplement 1A).

Coupling efficiency (%) is therefore the percentage of mitochondrial respiration allocated to produce ATP. Our results showed that coupling efficiency of WT-infected hMDMs is similar to Δ*dotA*- and non-infected hMDMs, or even increased. These results suggest that, even if basal respiration is highly reduced in WT-infected cells compared to Δ*dotA*- or non-infected hMDMs, the percentage of mitochondrial respiration allocated to produce ATP in WT-infected cells is not very different in all infection conditions. Accordingly, proton leak in WT-infected cells was minimal. Therefore, the results of coupling efficiency in WT-infected cells suggest that decreased ATP production observed in these WT-infected cells in our previous work (Escoll et al. Cell Host and Microbe 2017, Figure 5D) is caused by decreased basal respiration at 6 hpi (Escoll et al. Cell Host and Microbe 2017, Figure 5E, and this manuscript, Figure 1B), and not by a decreased efficiency in the coupling of mitochondrial respiration to ATP production in infected cells. The calculation of Coupling efficiency (%) has been added to the Methods section (Lines 483-488).

(3) I could not find "Nolfi-Donegan et al., 2020" (line 71) in Reference.

We are sorry, the reference is now added to the list of references (Line 639).

Significance:The authors previously reported that L. pneumophila reduced mitochondrial OXPHOS upon hMDMs infection depending on the T4SS function, revealing the pathogen-induced host cellular Warburg-like metabolism (Escoll et al., Cell Host and Microbe, 2017). This manuscript is a nice follow-up of the report by focusing on bacterial effector(s)-mediated conservation of the mitochondrial membrane potential, which can reflect the bacterial manipulation of the H^+^ circuit and the F0F1-ATPase activity.I feel enthusiastic about their finding that L. pneumophila can alter the mode of the mitochondrial ATP synthase by presumable function(s) of effector proteins. With the well-considered analytical data, the authors reasonably stated that the significance of the Legionella-induced "Reverse Mode" activity of the ATP synthase is for delaying cell death to preserve the bacterial survival niche.Overall, the manuscript is well-organized and can attract a broad audience not limited in the bacterial pathogenesis field.Referee Cross-commenting:I found that all the reviewers have provided reasonable suggestions and have properly evaluated the manuscript.Reviewer #3 (Evidence, reproducibility and clarity):The manuscript by Escoll and colleagues explores the complex interaction established between bacterial pathogens and host cell mitochondria during infections, using Legionella pneumophila as a model. This study builds on important advances of the team on this subject, as they previously reported how L. pneumophila effector proteins fragment the mitochondrial network at early time points of infection. Mitochondrial fragmentation and the accompanying reduction of oxidative phosphorylation (OXPHOS) normally lead to the loss of mitochondrial membrane potential and cell death. Surprisingly this is not observed in Lp-infected cells. Thus, the authors investigate whether Lp can interfere with these processes to preserve its replicative niche.They indeed observe that Lp-induced mitochondrial fragmentation is accompanied by a reduction in mitochondrial respiration. However, Lp seems to preserve mitochondrial membrane potential by inducing the reverse mode of the mitochondrial FoF1 ATPase, by a mechanism that depends on the activity of the Type 4 Secretion System (T4SS), suggesting involvement of bacterial effectors. Indeed, by screening Lp effectors targeting mitochondria, the authors show that LpSpl is partially involved in this process. Importantly, inhibition of the reverse mode of the mitochondrial ATPase is detrimental for Lp infections and trigger cell death. Correlative analysis of mitochondrial membrane potential, cell death and size of Legionella-containing vacuoles seems to indicate that Lp replicates preferentially in cells that preserve membrane potential and are thus protected from cell death.Major comments:The authors convincingly show how Lp induces the reverse mode of the mitochondrial FoF1 ATPase to preserve membrane potential during infection, which is a very important finding per se. However, the absence of a molecular mechanism underlying this phenomenon dampens the impact that the manuscript might have on the community.

As requested by Reviewers #1 and #2, we have performed experiments using the complemented strain (infection) and exogenously expressed *Lp*Spl effector (transfection) in order to strengthen the connection between *Lp*Spl and the effects observed in the F_O_F_1_-ATPase and cell death. These results are shown in new Figures 2I, Figure 2—figure supplement 1B to 1F, and Figure 4 —figure supplement 1E to 1G.

Thanks to these experiments performed during the review process of this manuscript we now have new tools to further investigate in detail the molecular mechanisms employed by LpSpl during infection. Indeed, we have provided first clues in new Figures S3F and S3G, where we showed that residue K366 is important for *Lp*Spl functions delaying host cell death. Transfection of a *Lp*Spl protein harboring a point mutation in this site (*K366A) had a reduced ability to protect transfected HEK-293 cells from STS-induced cell death compared to WT protein (new Figure S4B), highlighting the importance of this residue. We showed in a previous paper that the K366A mutation drastically decreased the activity of the enzyme (Rolando et al. Proc Natl Acad Sci USA 113(7):11901-6). According to the *Lp*Spl structure, this residue is solvent-exposed in the active site and belongs to the same loop as K353, which forms a Schiff base with pyridoxal 5′-phosphate (PLP), the cofactor that SPL uses to irreversibly degrade S1P into phosphoethanolamine and hexadecenal. In the human SPL structure, the equivalent residue to K366 (K359) is also solvent-exposed in the active site and is 7Å from the phosphate group of PLP. Due to its proximity to the PLP and putative substrate-binding region, it is thought that the *Lp*Spl K366A mutation affects interactions with the cofactor and/or substrate. Thus, our experiments revealed that the enzymatic activity of *Lp*Spl protein and its interaction with the PLP cofactor, which leads to the irreversible degradation of S1P, are imperative to protect transfected HEK-293 cells from STS-induced cell death.

In figure 1 The TMRM approach convincingly shows that there is no significant loss of mitochondrial membrane potential between non-infected, WT Lp-infected and DdotA-infected cells. The charts presented in figure 1D and E would greatly benefit from the integration of an additional condition where mitochondrial fragmentation is induced in non-infected cells to demonstrate the efficiency of the TMRM approach in monitoring loss-of-potential.

As requested, we have performed these experiments (also asked for by Reviewer #2) in hMDMs isolated from 3 different donors. They confirmed our results (new Figure 1E and Figure 1 —figure supplement 1D). Our results in hMDMs isolated from Donor #2 using FCCP to completely depolarize mitochondria have been included in Figure 1E, and the entire single-cell dataset from the 3 donors used for these experiments has been added to the supplementary Figure S1 as new panel D.

We have updated the text in Lines 159-164, it reads now:

“Single-cell analyses (Figure 1E, Figure 1—figure supplement 1D) showed that Lpp-WT-, Lpp-∆dotA- and non-infected single hMDMs showed a wide range of Δψ_m_ values at any time-point (Figure S1C) with no significant differences between Lpp-WT- and Lpp-∆dotA-infected hMDMs at 6 hpi. Infected cells with both strains showed a significantly higher Δψ_m_ compared to non-infected cells (p<0.0001, Figure 1E, Figure 1—figure supplement 1D).”

On line 240 of page 8, the authors state that OXPHOS cessation and collapse of membrane potential can trigger cell death. Given that this is at the basis of the proposed model, it would be important to test this hypothesis in non-infected cells and compare the percentage of living cells with infected cells. This would also help the reader interpreting the data presented in chart 4A as the variations in living cells are minimal (albeit significant) among the conditions.

We have performed these experiments in non-infected cells by adding FCCP to fully collapse the Δ*ψm*, and then we have compared the % of living cells to non-treated cells (new Figure 4—figure supplement 1A). Our results show that the FCCP-induced collapse of the Δ*ψm* decreased the % of living cells (Annexin-V ^neg^ cells), as expected. These results, shown in new Figure S3A, now provide reference values to interpret the data shown in Figure 4A, as the Reviewer suggested. Results of % of living cells obtained in infected cells (Figure 4A) can be now compared to intact non-infected cells as well as to non-infected cells with fully collapsed Δ*ψm* (Figure 4—figure supplement 1A).

These results are added in Line 281-284, it reads now:

“To test this hypothesis, we first measured the percentage of living cells after a FCCP treatment of 18 h. An FCCP-induced collapse of the Δψm reduced the percentage of living cells by 32% compared to non-treated hMDMs (p = 0.0094, Figure 4—figure supplement 1A).”

Using the size of bacterial vacuole as a proxy for bacterial replication in charts 4F and G seems like an odd choice as vacuole size is not necessarily indicative of bacterial replication. Measuring the size of intracellular bacterial "colonies" or, even better, their fluorescence intensity using fluorescently-tagged Lp would be a far better option.

As we measured the area (μm^2^) of GFP bacteria inside infected cells in Figures 4F and 4G, in order to avoid confusion and in agreement with Reviewer’s suggestion, we changed the term “*bacterial vacuole*” by the term “the area of GFP expressing bacteria, a proxy for intracellular replication”, which was increased in infected cells when 12 hpi and 18 hpi were compared (Figures 4F and 4G). Accordingly, the main text and the legend of Figure 4 were changed as follows:

Line 332-338 now reads:

“Furthermore we correlated the area of GFP expressing bacteria, a proxy for intracellular replication, with the Δψm and cell death at the single-cell level (12 and 18 hpi, Figure 4F and 4G respectively). […] This indicated that conservation of Δψm and delayed cell death are required for bacterial replication; as they prolong macrophage survival.”

Line 778-784 now reads:

“(F) Same infection conditions as in (E) but using non-treated cells, and bacterial replication was monitored in each Lpp-WT-infected single cell. Single-cell analyses (12 hpi) of Δψm (TMRM SD/Mean), area of intracellular GFP-expressing bacteria (μm^2^), a proxy for intracellular replication, and cell death (Annexin-V intensity) in more than 3800 cells are shown. Single-cell data from one representative experiment; Color scale (yellow) represents Annexin V intensity per cell (AU). (G) Same as in (F) at 18 hpi.”

Minor comments:Overlaying median and error bars on top of the shapes in figure 2H would facilitate data analysis.

This was changed as requested. The error bars are on top of the shapes in Figure 2H.

There is a typo on line 75 of page 3 (“reveito”).

Fixed.

Line 211 and 213 page 7: the DdotA mutant is not "lacking" the T4SS, it is just defective.

Changed. Line 214 now reads:

“To learn if any of these effectors is involved in the T4SS-dependent induction of the “reverse mode” of the F_O_F_1_-ATPase, we infected hMDMs for 6 hours with Lpp-WT or its isogenic mutants lacking a functional T4SS (Lpp-∆dotA)”

As the chart in figure 4A illustrates the percentage of living cells among infected cells, it would seem more intuitive to indicate % of living cells for the y axis.

Changed. Now the Y axis of Figure 4A indicates “% of living infected cells”.

It would be useful to indicate the full name of TMRM somewhere in the manuscript.

As requested, we indicated the full name. Line 151 it now reads:

“We developed a miniaturized high-content assay based on kinetic measurements of Tetramethylrhodamine Methyl Ester (TMRM) fluorescence in non-quenching conditions (10nM)”

Significance:This is a very well presented study adding important information on a very timely subject, which is the interplay between bacterial pathogens and host mitochondria. In addition, the work presented makes use of microscopy-based approaches allowing single-cell analysis, which is very important to specifically investigate infected cells.Referee Cross-commenting:All three reviews agree on several key points of the manuscript. I have no additional comments.